# Intriguing Properties of Quantization at Scale

**Arash Ahmadian** [*][†]
Cohere For AI
arash@cohere.com

**Saurabh Dash** [*]
Cohere
saurabh@cohere.com

**Hongyu Chen** [*]
Cohere
charlie@cohere.com

**Bharat Venkitesh**
Cohere
bharat@cohere.com

**Stephen Gou**
Cohere
stephen@cohere.com

**Phil Blunsom**
Cohere
phil@cohere.com

**Ahmet Üstün**
Cohere For AI
ahmet@cohere.com

**Sara Hooker**
Cohere For AI
sarahooker@cohere.com

## Abstract

Emergent properties have been widely adopted as a term to describe behavior not present in smaller models but observed in larger models (Wei et al., 2022a). Recent work suggests that the trade-off incurred by quantization is also an emergent property, with sharp drops in performance in models over 6B parameters. In this work, we ask *are quantization cliffs in performance solely a factor of scale?* Against a backdrop of increased research focus on why certain emergent properties surface at scale, this work provides a useful counter-example. We posit that it is possible to optimize for a quantization friendly training recipe that suppresses large activation magnitude outliers. Here, we find that outlier dimensions are not an inherent product of scale, but rather sensitive to the optimization conditions present during pre-training. This both opens up directions for more efficient quantization, and poses the question of whether other emergent properties are inherent or can be altered and conditioned by optimization and architecture design choices. We successfully quantize models ranging in size from 410M to 52B with minimal degradation in performance.

## 1 Introduction

The push for ever larger language models (LLMs) has been driven by a strong correlation between performance and the number of parameters (Chowdhery et al., 2022; Zhang et al., 2022; Kaplan et al., 2020). This has led to new breakthroughs in downstream performance, but has also posed new challenges in making these models accessible. Larger models incur higher memory and latency because of the requirement to store many more model weights and the optimizer in fixed memory (Dehghani et al., 2021; Treviso et al., 2022). Due to the massive size of state-of-art LLMs, inference often requires hosting across multiple machines which limits the practical usability of such models.

To address this, much research has focused on compression techniques such as quantization, which reduces the number of bits needed to represent each learned parameter in a model (Gholami et al., 2021). Quantization techniques are widely used in smaller model regimes – quantizing weights

---

[*]Equal Contribution

[†]Also affiliated with the University of Toronto & the Vector Institute for Artificial Intelligence.

37th Conference on Neural Information Processing Systems (NeurIPS 2023).

stored as 32-bit or 16-bit floating-point numbers to 8-bit integers (`INT8`) produces large reductions in memory and latency.

However, at scale simple quantization techniques have been shown to lead to a pronounced degradation in performance (Xiao et al., 2022). This trade-off has been attributed by several recent works (Dettmers et al., 2022; Zeng et al., 2022; Xiao et al., 2022; Bondarenko et al., 2021) to *emergent outlier dimensions*—scaling transformer-based architecture results in large activation magnitude outliers which are concentrated along a few hidden dimensions. To remedy this degradation, mixed-precision solutions have been proposed that handle the outlier activations separately (Dettmers et al., 2022). While effective at preventing degradation, these specialized techniques pose significant latency overhead which negates some of the benefits to memory (Wu et al., 2020). In Appendix C.1.3, we demonstrated the efficiency benefits of using simpler quantization techniques over mixed-precision solutions.

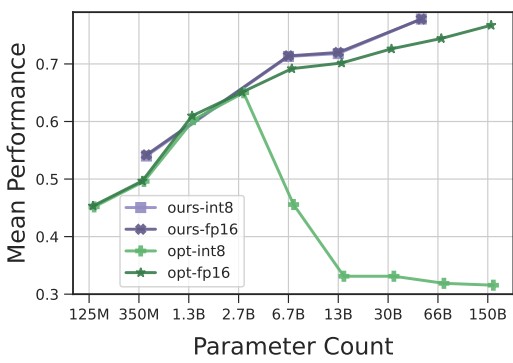

Figure 1: Mean zero-shot accuracy on HellaSwag, PIQA, LAMBADA, and WinoGrad. In contrast to the OPT family, our models show minimal degradation after simple vectorwise quantization. Data points for OPT models are from (Dettmers et al., 2022).

The difficulties of quantizing at scale prompt us to ask: *are emergent properties due to nature or nurture?* Recent work introduces intriguing and somewhat contradictory answers to this question: Models like OPT-175B (Zhang et al., 2022) and FairSeq (Artetxe et al., 2022) exhibit pronounced sensitivity to post-training quantization and require complex mixed-precision decomposition quantization methods (Dettmers et al., 2022; Wei et al., 2022b; Bondarenko et al., 2021; Luo et al., 2020; Zeng et al., 2022). On the contrary, BLOOM-176B (Scao et al., 2022) is easier to quantize with a simple quantization recipe and a relatively small performance drop (Frantar et al., 2022; Xiao et al., 2022). Zeng et al. (2022) hypothesize that the observed difference in weight distribution characteristics may be due to the difference in optimization choices made during pre-training.

In this work, we seek to reconcile these observations. We posit that it is possible to optimize for a quantization friendly training recipe that suppresses large activation magnitude outliers. This leads to a distribution of activations and weights that are more amenable to simple INT8 quantization recipes and does not necessitate the need for complex and inefficient mixed-precision computations. Our results show that we can introduce simple INT8 post-training quantization with negligible impact on performance due to choices we make during the pre-training stage. As shown in Figure 1, across 8 zero-shot downstream tasks, our models do not present any significant performance drop, having only 0.24% average degradation in a 52 billion parameter model.

In summary, our contributions are as follows:

- We conduct a controlled large scale study – At 6B, we maintain the same architecture and vary key optimization choices such as weight decay, gradient clipping, dropout and precision of training representation. We present results with optimal hyper-parameters across models varying from 410 million to 52 billion parameters, with each experiment variant trained from random initialization. While this requires a compute intensive set-up, it allows us to rigorously disentangle what factors actually influence sensitivity to quantization.

- We show that reoccurring activation outliers are not a universal *emergent property* of LLMs at scale and can be avoided at scales as large as 52B given the right optimization choices. Our 52B parameter model shows only 0.26% performance degradation across 8 tasks with INT8 PTQ quantization of *both* activations and weights, which enables a 1.4-1.5x inference speedup and around 40% memory footprint reduction over half-precision inference.

- We contribute a fine-grained analysis of activations and weights, and show that several key weight and activation characteristics may explain the difference in sensitivity between our robust models and models like OPT which have been shown to have pronounced sensitivity

to quantitization at scale. We hope these insights help guide future model design and pre-training strategies.

## 2 Background

Quantization refers to compressing weights and activations of a neural network into lower-bit representations. Here, our focus is on **one-shot post-training quantization** (PTQ) (Xiao et al., 2022; Dettmers et al., 2022), which quantizes the network post-training without additional finetuning steps for calibration. Given the complexities of successfully training a large language model (Zhang et al., 2022; Rae et al., 2021), PTQ methods are extremely attractive as these techniques require the least modification to pre-trained parameters, compared to quantization-aware training methods (Zafrir et al., 2019; Krishnamoorthi, 2018) or quantization-aware finetuning (Park et al., 2022b; Yao et al., 2022; Frantar et al., 2022; Zhuo et al., 2022; Li et al., 2021; Hubara et al., 2020; Nagel et al., 2020) which both require updates to model weights. We include more detail about each of these broad groups of techniques in Appendix C.

To-date PTQ techniques that quantize both the activations and weights of a language model have proven extremely challenging at large scale (>6B parameters) leading to pronounced drops in performance. Instead, less aggressive techniques have been used such as *weight-only quantization* (Gerganov, 2023; Frantar et al., 2022; Zeng et al., 2022) that leaves the activations in higher precision or *quantization with mixed-precision decomposition* (Dettmers et al., 2022) which decomposes the matrix multiplication to compute a small fraction of elements at a higher precision (FP16) while the bulk of the computations is performed at low precision (INT8).

*Weight-only quantization* brings speedup to inference by reducing the amount of data movement. However, as large language models are scaled, progressively they become compute-bound and the improvements due to weight-only quantization stagnate. While *mixed-precision decomposition* approaches have theoretical latency benefits due to the bulk of the computation being performed at lower precision, in practice without specialized hardware (Dash et al., 2022; Dash & Mukhopadhyay, 2020; Hooker, 2021), GPU kernels, or additional kernel calls to prepare the inputs and weights for mixed-precision computation, the projected benefits cannot be realized (Dettmers et al., 2022). To further realize latency gains, we need to quantize both *weights and activations* into 8-bit integers (INT8) to utilize specialized integer INT8 GEMM kernels, which are supported by a wide range of hardware (e.g., NVIDIA GPUs, Intel CPUs, Qualcomm DSPs, etc.). In Appendix C.1.3, we showed a 1.4x inference speedup using weight and activation quantization, compared to half-precision baseline. In addition, weight and activations quantization further enables compressing key-value cache to INT8. Since key-value cache takes up a significant part of the GPU memory during inference (Sheng et al., 2023), weight and activations quantization further contributes to memory saving and high throughput inference.

Hence, the most challenging *weights and activations* setting is the focus of this work. More concretely, given a neural network that learns a function $f$ parameterized by a set of weights $\{\mathbf{W_0}, \mathbf{W_1}, ..., \mathbf{W_n}\}$ with corresponding activations $\{\mathbf{X_0}, \mathbf{X_1}, ..., \mathbf{X_n}\}$, during quantization, activation and weight tensors denoted by $\mathbf{X} \in \mathbb{R}^{t \times h}$ and $\mathbf{W} \in \mathbb{R}^{h \times o}$ – where $t$ denotes the sequence-length, $h$ the input hidden units and $o$ the output hidden units[3] – are replaced with lower-bit counterparts $\mathbf{W_Q}$ and $\mathbf{X_Q}$ by scaling and rounding the original tensors to INT8 range. We focus on *vector-wise quantization* recipe (Dettmers et al., 2022) to increase quantization granularity. In vector-wise quantization, we define the row-scaling vector $\mathbf{s_x} \in \mathbb{R}^t$ and column-scaling vector $\mathbf{s_w} \in \mathbb{R}^o$ by calculating the scaling constants for each row/column through uniform symmetric mapping (Nagel et al., 2021). After INT8 matrix multiplication, dequantization of $\mathbf{X_Q}\mathbf{W_Q}$ back to FP16 is through element-wise multiplication with $\mathbf{s_x}$ and $\mathbf{s_w}$:

$$\mathbf{XW} \approx \mathbf{s_x} \odot (\mathbf{X_Q}\mathbf{W_Q}) \odot \mathbf{s_w} \tag{1}$$

Where $\odot$ denotes broadcastable matrix multiplication. The quantization and dequantization steps for the above do not add much memory overhead compared to quantizing with a single scaling constant for each weight or activation tensor, while significantly increasing the representation power of INT8.

---

[3]We omit the batch dimension for simplicity.

# 3 Methodology and Experimental Setup

## 3.1 Methodology

Our goal is to understand whether sensitivity to widely used quantization techniques is inherently an emergent property at scale or due to optimization choices made during pre-training. Recent work has presented seemingly contradictory empirical findings – some models such as OPT-175B show pronounced sensitivity at scale to post-training quantization while other models such as BLOOM-176B are relatively robust to post-training quantization.

| Experimental Axes | Choices |
|---|---|
| Weight decay | 0.001, 0.01, 0.1 |
| Gradient clipping | None, 1 |
| Dropout | 0, 0.1, 0.4, 0.8 |
| Half-precision | `bf16`, `fp16` |

Table 1: Optimization choices that are explored for pre-training in our controlled setup.

These models differ in numerous ways such as architectural differences, pre-training data, training infrastructure, and finally optimization choices,making it challenging to attribute differences in quantization performance. To rigorously isolate what choices result in sensitivity to quantization, we measure the impact of optimization choices within a tightly controlled experimental setup – training the same large scale model architecture from random initialization while rigorously varying only key aspects of the optimization procedure. Each optimization choice is evaluated in two ways: we measure the resulting degradation after PTQ in zero-shot downstream performance and then analyze the model weights and feature activations to understand how the characteristics at scale impact quantization performance.

Training multiple multi-billion parameter size language models is extremely expensive – a single 52B language model takes roughly 20 days of training with 2048 TPU cores.[4] Therefore, we first conduct our controlled experiments on 410M and 6B models using early checkpoints and then validate the results at scale, by fully training 6B, 13B, and 52B parameter size models with our most quantization friendly training recipe. In practice, we found performance at early checkpoints predictive of fully trained model performance.

We briefly describe each of the axes of variations below:

**Weight decay** Weight decay is widely used to impede over-fitting by penalizing large magnitude weights (Goodfellow et al., 2016). We experiment with a range of weight decay values {0.001, 0.01, 0.1}.

**Gradient clipping** Gradient clipping rescales the norm of the gradient vector if it exceeds the threshold (Pascanu et al., 2013). It is widely used in LLMs to prevent exploding gradients and accelerate training convergence (Du et al., 2021; Zhang et al., 2022). We experiment with a gradient norm threshold of 1 as well as training without gradient clipping.

**Dropout** Dropout is a widely used regularization technique that drops neurons with a probability of $p$ during training (Srivastava et al., 2014; Hinton et al., 2012). We apply dropout to the output of the self-attention block and the feed-forward layer before the corresponding residual connection as described in Vaswani et al. (2017), but we do not use a dropout for the input embeddings. We experiment with {0, 0.1, 0.4, 0.8} dropout probabilities.

**Half-precision data type: bf16 vs fp16** Training neural networks in mixed-precision is a common technique to reduce the memory requirement and improves training time while often achieving comparable performance to full-precision training (Micikevicius et al., 2017). In this technique, a copy of weights is stored in full-precision (`fp32`) whereas the forward and backward passes are done using half-precision in either float16 (`fp16`) or bfloat16 (`bf16`) (Kalamkar et al., 2019; Dean et al., 2012; Abadi et al., 2015).

We experiment with `fp16` and `bf16`. Furthermore, for each half-precision data type, we vary weight decay values of (0.1, 0.01) to observe whether the effect of the half-precision data type is exasperated with a smaller weight decay value of 0.01.

---

[4]We include more details about the hardware and training requirements in Section 3.2

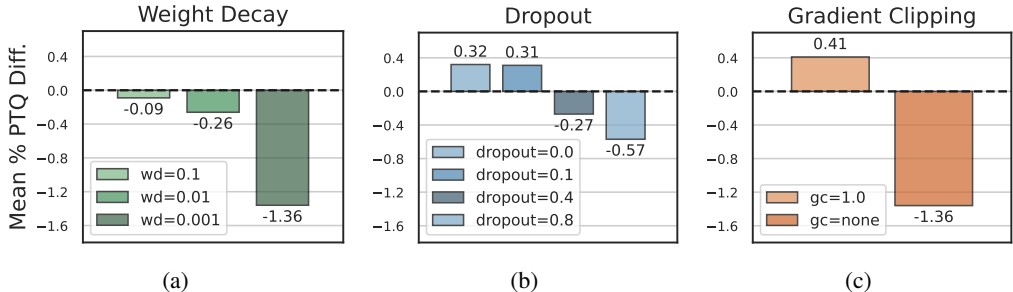

Figure 2: Study of the PTQ performance when varying weight decay, dropout, and gradient clipping. In Figure 2c a control weight decay value of 0.001 is used to to minimize the effects of weight decay when studying gradient clipping. Otherwise, we use weight decay of 0.1, dropout of 0, and gradient clipping threshold of 1.0 as control variables.

## 3.2 Experimental Setup

**Model** We train autoregressive decoder-only Transformer models (Liu et al., 2018) with a standard language modeling objective. Given an input sequence of $S = [s_1, \cdots, s_t]$, a language model with parameters $\theta$ trained to minimizes the following negative log likelihood:

$$L(S) = \sum_i -logP(s_i|s_{<i}; \theta) \tag{2}$$

Our language models follow the traditional GPT style architecture reported in Radford et al.. Different from the Radford et al., we do not share the same weight matrix for input and output embedding layers. Instead, we learn separate input and output projections to enable higher model parallelism. We train models with parameter sizes ranging from 410 million to 52 billion. All models have a maximum sequence length of 2048 tokens. We use SentencePiece (Kudo & Richardson, 2018) tokenizer with a vocabulary of 51200 to tokenize the text.

**Training details** We pre-train models using a mixture of datasets from Common Crawl and C4 (Raffel et al., 2020) with AdamW (Loshchilov & Hutter, 2019) optimizer and a batch size of 256. We use a cosine learning rate scheduler with 1500 warm-up steps. We use GeLU activations (Hendrycks & Gimpel, 2016). All the models are trained with mixed-precision, i.e. forward and backward passes are computed in `bf16` or `fp16` half-precision format, but the model parameters are stored in `fp32` in the distributed optimizer state (Rajbhandari et al., 2020). For half-precision `fp16`, we experimented with a variant by switching layernorm arithmetic from `fp16` to `fp32` for better numeric stability (Micikevicius et al., 2017). Without layernorm arithmetic being in `fp32`, the training was unable to converge.

To avoid an exorbitant computational cost given each variant requires training from scratch and we are evaluating very large scale models, we first iterated on 410M models but observed very minimal degradation. As a result, we scaled to 6B parameters which is the scale at which we present our results. Following the analysis, we validate our findings by scaling the most PTQ friendly optimization choices to 13B and 52B models trained to more than 200k steps. We note that the 52B model required training time of roughly 20 days and in total we trained 17 models of sizes ranging from 410M to 52B to carefully measure the impact of each ablation. The details of each model's architecture and training hyper-parameters are provided in Appendix A.1.

**Infrastructure** We use TPU-v4 chips (Jouppi et al., 2017) to train, and Nvidia A100 GPUs to evaluate our models. All models are trained using the `FAX` (Yoo et al., 2022) framework which enables efficient model and data parallelism. It takes approximately 72 hours on 128 cores to train a 6B parameter model for 75000 steps.

**Evaluation** We evaluate each model variant on Copa (test and dev set) (Wang et al., 2019), HellaSwag (Zellers et al., 2019), PIQAValidation (Bisk et al., 2020), StoryCloze (Mostafazadeh et al., 2016), WinoGrande (Sakaguchi et al., 2019), Paralex (Fader et al., 2013), and LAMBADA (Paperno et al., 2016). All evaluations were done in a zero-shot setting. In total, we benchmark on 8 tasks comprised of Multiple choice (MC) completion, MC Co-referencing, Generation, and Question Answering (QA) types. Details of our evaluation suite are given in Appendix A.2. For each experimental variant, we

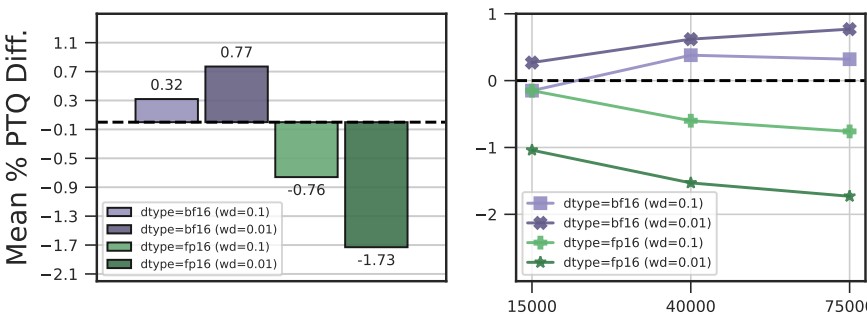

Figure 3: Study of PTQ performance when varying the precision used during training. On the left, the performance difference is plotted at 75000 steps whereas on the right, it is plotted over time. We observe that `fp16` training consistently leads to models which are far more sensitive to post-training quantization.

report the average percent performance difference and the pre-quantization and post-quantization performances across tasks. The percent degradation is calculated by normalizing each task's absolute degradation by the corresponding pre-quantization performance.

## 4 Results and Discussion

For each experimental axis, we train the corresponding variants to a maximum of 75000 steps. Note that due to the enormous computational cost of full-pretraining of these models, we only fully-train the final variants with optimal hyper-parameters (for >200k steps). Below we present a breakdown of the degradation results and analysis for each experimental axis. All variants with the exception of `dropout=0.8`, had similar pre-quantization performance. This is important, as we are interested in comparing optimization choices that still result in models of comparable quality, but differing sensitivities to post-training quantization. Refer to Appendix A.3 for the per-task breakdown of results.

**Weight Decay** As can be seen in Figure 2a, we observe that a higher level of weight decay during pre-training improves post-training quantization performance. We do not use gradient clipping in these experiments to isolate the impact of weight decay. A larger weight decay value (`0.1` vs `0.001`) results in better post-training performance (0.09% vs 1.36% degradation). Furthermore, as shown in Figure 3, combining lower weight decay with `fp16` can further amplify sensitivity to post-training quantization. A small weight decay value (0.01) can cause higher performance degradation in post-quantization after training (1.73%).

**Dropout and Gradient Clipping** In Figure 2b we observe that higher levels of dropout correspond to sharper degradation in post-training quantization. Note that $P_{dropout} = 0.8$ unsurprisingly leads to a poor absolute downstream performance, however, it helps establish a clear trend given other data points. Figure 2c shows the relative quantization degradation for models with and without gradient clipping. When varying gradient clipping, a control weight decay value of 0.001 is used to minimize the impact of weight decay. As seen in the figure, gradient clipping shows a positive impact on the quantization performance, improving robustness to post-training quantization. This suggests that gradient clipping to an extent counteracts the effects of a small weight-decay value which would otherwise lead to higher quantization degradation.

**Half-precision: bf16 vs fp16** Figure 3 shows the quantization degradation and absolute performance for `fp16` and `bf16` for 6B parameter models. Training with `fp16` leads to higher quantization degradation than `bf16`. We relate the degradation results to the numerical stability in training. `fp16` format uses a smaller range for the exponent than `bf16`. While `bf16` uses 8 bits for the exponent, `fp16` only uses 5. Most floating point formats also have denormalized numbers which allow for a soft underflow. This can get exponentially closer to 0.0f for each additional bit in the mantissa. This makes underflow more of a concern for floating point formats.

**Positional Embeddings** Since choice of positional embeddings is often a differentiating factor between architectures e.g. OPT models use learned embeddings (same as us) whereas the BLOOM family uses AliBi Press et al. (2022), we also trained 3 models using learned and sinusoidal positional embeddings, and ALiBi with otherwise optimal hyper-parameters (`bf16` training, `weight-decay=0.1`, `dropout=0.0`, `gradient clipping=1.0`. We did not see a notable difference in downstream PTQ degradation as all values were within 0.06% of another.

Notably, we observe that our findings provide insights into the quantization robustness of BLOOM-176B, which to our knowledge is the only open-source LLM (with more than 50B parameters) and a decoder-only block architecture, that is trained with `bf16`; compared to simliar sized `fp16` trained models such as OPT-175B (Xiao et al., 2022).

**Using Early Checkpoints to Infer Converged Model Behavior** Given the considerable computational cost of our experiments, it is valuable to explore whether converged model behavior can be inferred from checkpoints from early snapshots of training. In Figure 3, we plot the relative post-training quantization degradation given checkpoints trained with different levels of precision at different steps during pre-training. We observe that quantization degradation increases with step count, but the relative impact of varying the bit representation emerges early in training which confirms the main trend. Interestingly, `fp16 (wd=0.01)` variant exhibits high quantization degradation in the starting phase training as early as 15000 steps.

**Outlier Analysis** As a potential proxy measure of PTQ degradation, we measure the number of *outlier dimensions* as it's been shown to strongly effect activation quantization sensitivity of LLMs(Dettmers et al., 2022). Activation feature dimensions are classified as outlier dimensions when the activation values (denoted as $\alpha$) are greater than 6.0 at more than 25% of layers and 6% of tokens. However, we find that a threshold of 6.0 is too high to classify a feature dimension as an outlier for all the variants we consider. After correspondence with the authors, we also explored various adaptations of this outlier detection recipe presented in Dettmers et al. (2022) to make it generalizable. However, we did not observe a clear correlation between these measures and sensitivity to quantization. We refer to Appendix B.3 for detailed treatment of these replication efforts.

**Scaling Insights to 52B scale** To validate our experimental findings at scale and with fully trained models, we pre-train 410M, 6B, 13B, and 52B parameter models for more than 200k steps, using the best optimization choices with respect to robustness in post-training quantization: weight decay of 0.1, no dropout, gradient clipping of 1, and `bf16` as the half-precision format. Figure 1 shows mean zero-shot accuracy for the non-quantized and quantized model using INT8 weights and activations. Compared with OPT models (Zhang et al., 2022), our fully-trained models are significantly more robust to post-training quantization starting from 6B parameter size. Our largest scale model with 52B parameters, even shows a **0.08%** average *improvement* across the evaluation suite and **0%** degradation across LAMBADA, HellaSwag, and PIQA where OPT-66B which is the closest OPT model in terms of size, has an extreme drop in performance of 42% as reported in Dettmers et al. (2022).

**Transferability of Results** All the models that we experimented with are architecturally very similar to the OPT & BLOOM family as they are both decoder-only, pre-layernorm (where the layernorm is placed before the linear layers), and use vanilla attention. They differ in the number of blocks and the hidden dimensions of models at different sizes. Since we thoroughly experimented with different model sizes, we believe that our insights directly translate to the OPT& BLOOM families' architectures.

To evaluate our models' performance under a different quantization recipe in addition to INT8, we also test 4-bit integer (INT4) column-wise weight-only quantization, similar to Du et al. (2021). Our 52B parameter model exhibited only a 3.6% relative drop in mean zero-shot performance across the 8 evaluation tasks. It is worth noting that this quantization scheme does not require any fine-tuning or optimization and hence these results highlight the high robustness of our trained models. In comparison, when applying the same quantization scheme to BLOOM, BLOOM-176B and BLOOM-7B show 29.5% and 18.7% degradation respectively on LAMBADA as reported in the GLM-130B technical report (Du et al., 2021), while our 52B model only has 8.6% degradation.

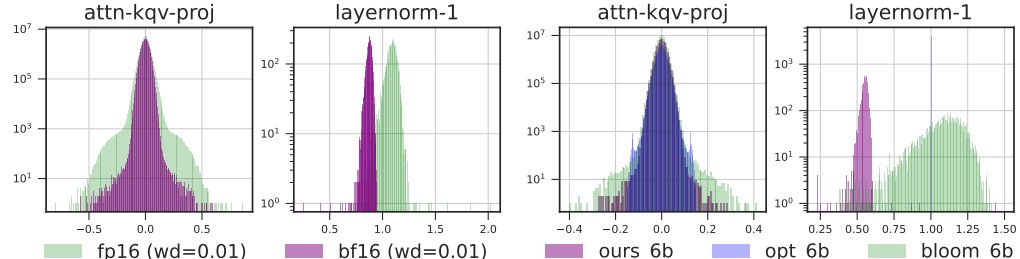

Figure 4: Weight distribution of `attn-kqv-proj` and `layernorm` gain (**g**) parameter in an example block (Block 14) for both `fp16`/`bf16` variants and our final 6B model in comparison with OPT and BLOOM. Weight distributions for all blocks are shown in Appendix B.1

## 5 Weight and Activation Analysis

Our results in Section 4 find that sensitivity to quantization at scale is not an inherent emergent property. Rather, it can be avoided at scales as large as 52B given the right optimization choices. We also found that outlier dimensions are not always a consistent proxy for quantization sensitivity. Thus as an attempt to address the short-comings of outlier dimensions as a proxy and to reconcile our observations on degradation trends, we perform a fine-grained analysis of activations and weights to understand how the trained distribution of our models differs from models like OPT that are far more sensitive to quantization. we hope that the metrics we propose and evaluate in this section, help further discussion about useful proxy metrics for guiding pre-training optimization choices to improve robustness to quantization. For all the metrics proposed below, we include the complete analysis for all layers in Appendix B.

**Activations** As a first step, we analyze input activations of the attention projection (`attn-kqv-proj`) as it is the earliest point for INT8 multiplication in a decoder block. Here, we measure root-mean-square error $RMSE(\mathbf{X}, \hat{\mathbf{X}})$ where $\hat{\mathbf{X}}$ denotes the de-quantized activations. Additionally, we report the mean standard deviation of the input activations measured per token. While $RMSE(\mathbf{X}, \hat{\mathbf{X}})$ directly indicates the quantization error, standard deviation (STD) has been shown to be closely related to the expected quantization error of a normally distributed input (Kuzmin et al., 2022). Figure 5 compares the `bf16` and `fp16` variants. We observe that the RMSE and STD of `fp16` are far higher than the `bf16` variant – the RMSE for the `fp16` variant is 6.9x the RMSE for the `bf16` variant. This difference is even more pronounced if we compare our model to the OPT: the RMSE and STD of the OPT are 27.7x and 1.8x higher respectively relative to our model (Figure 5; Bottom row).

**LayerNorm** Since we use a *pre-norm* architecture, the input activations to `attn-kqv-proj` and `outer-expander-mlp` are both preceded by a layernorm. The layernorm gain parameter, $\mathbf{g} \in \mathbb{R}^h$, directly influences the output activation tokens' spread, and can significantly vary in distribution shape as seen in Figure 4. We generally observe that within our experimental axes, the standard deviation of the gain parameters for both the first and second layernorms are higher in a significant number of layers for variants with higher degradation compared to others in the same axis. In Figure 5, we compare the standard deviation of **g** for self-attention layernorm and we observe that STD(**g**) is 2x higher for the `fp16` variant relative to `bf16`. Our findings add further support to previous work which suggests that the gain parameters act as an outlier amplifier and thus further quantization degradation through scaling (Wei et al., 2022b).

In the bottom row of Figure 5, we also compare STD(**g**) of our model relative to OPT and BLOOM. We observe that even the BLOOM model that is relatively robust to quantization has a far larger STD(**g**) than our model with a multiplier of 5x. Interestingly, we find that OPT-6B layernorm gain parameters are all set to 1.0 while biases varied as expected. Hence, given the gain parameters appear to be hardcoded, the STD(**g**) of the OPT model is 0. We were not able to find any mention of such design decision either in Zhang et al. (2022) or the github repository: `https://github.com/facebookresearch/metaseq`.

**Attention Projection Weights** Finally, we compare the weight distribution of `attn-kqv-proj` layers. As seen in Figure 4, the `fp16` variant has a significantly wider distribution compared to `bf16`.

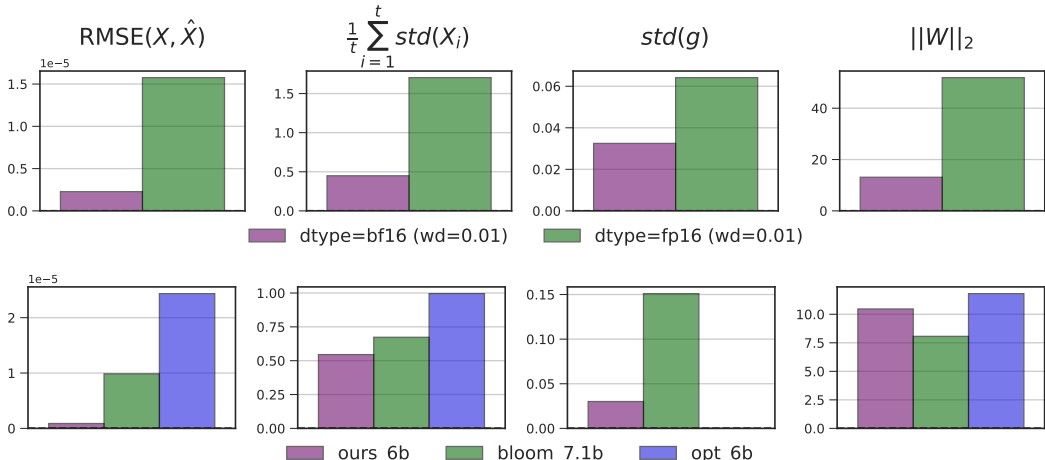

Figure 5: Average input token activation STD, preceding layernorm gain STD, and spectral norm of linear weights, follow trends similar to those observed in $\text{RMSE}(\mathbf{X}, \hat{\mathbf{X}})$. Plots correspond to the `attn-kqv-proj` layer in an example block (Block 14). Comparisons for all other blocks are given in Appendix B.2. **Top row:** Comparison of `fp16` and `bf16` variants. **Bottom row:** Our 6B model trained with optimal hyper-parameters compared against similar sized OPT-6B and BLOOM-7.1B models

Additionally, inspired by Lin et al. (2019), we use spectral norm to measure the maximum degree of noise amplification for each token activation.

For a given weight matrix $\mathbf{W} \in \mathbb{R}^{h \times o}$, and input token activation noise $\mathbf{x}_\delta \in \mathbb{R}^{\mathbf{h}}$, the spectral norm $\|\cdot\|_2$ is defined as

$$\|\mathbf{W}\|_2 = \sup_{\mathbf{x}_\delta \neq 0} \frac{\|\mathbf{W}\mathbf{x}_\delta\|_2}{\|\mathbf{x}_\delta\|_2} = \sigma_{max} \tag{3}$$

where $\sigma_{max}$ is the largest singular value of $\mathbf{W}$.

As seen in Figure 5, we observe that the spectral norm of the `fp16` variant is 4x higher than the `bf16`. In addition, on the bottom row of Figure 5, we observe that both BLOOM and our model have generally lower spectral norm than OPT 6B that is far more sensitive to quantization.

## 6 Related Work

**Challenges of Quantization at Scale** Recently, there have been several studies to characterize the emergence of outliers at scale, and relate this to the difficulties in post-training quantization of both weights and activations (Dettmers et al., 2022; Wei et al., 2022b; Puccetti et al., 2022). Dettmers et al. (2022) depict a phenomenon of *emerging outliers* by observing that large outlier dimensions systematically emerge at a certain scale (6.7B parameters) which hamper quantization attempts. Extreme outliers at scale was also empirically confirmed in follow-up works (Zeng et al., 2022; Xiao et al., 2022). The causes of outliers have also been the subject of recent work. Puccetti et al. (2022) observe that in Masked Language Models (MLMs) the magnitude of hidden state coefficients corresponding to outlier dimensions correlates with the frequency of encoded tokens in pre-training data. Wei et al. (2022b) observe that LayerNorm scaling (**g**) amplifies the outliers and can be suppressed using a modified LayerNorm and token-wise clipping. Wortsman et al. (2023) consider large-scale vision-language models and show that quantization techniques are more stable if the network is trained and initialized so that large feature magnitudes are discouraged.

Most mitigation strategies to quantize in the presence of outliers has required more complex quantization techniques. For example, Dettmers et al. (2022) propose selective mixed-precision computation by only computing the outliers at higher precision. However, such a setup proves difficult to map to hardware, limiting the inference speedup. Xiao et al. (2022) propose to smoothen out these outliers by migrating some of the activation variances into the model weights with appropriate scaling.

Although the authors demonstrate the ability of this framework to scale to large models, additional rescaling is required for activations which leads to additional latency overhead without specialized kernels. Another limitation of Xiao et al. (2022) is that it relies on the assumption that outliers exist in activations, and that weights can bear additional outliers and still be easy to quantize.

Our work is the first to show that outliers are not inherent to scaling large language models. Rather than an emerging property, they are a result of particular training methods. Compared to previous methods using extensive quantization schemes with custom kernels (Dettmers et al., 2022), our work applies PTQ using simple, one-shot linear weight and activation quantizations which can take advantage of NVIDIA-provided CUTLASS kernels, leading to a significant decrease in latency and memory footprint (Table 6).

# 7  Conclusion

We present a rigorous study of the effect of how various optimization choices affect INT8 PTQ with the goal of reconciling the recent contradictory observations regarding emergent properties in Large Language Models. We show that regularization directly impacts PTQ performance and that higher levels of regularization through common techniques such as weight-decay, and gradient-clipping leads to lower post-training quantization degradation. We further demonstrate that the choice of half-precision training data type has a significant impact on PTQ performance – emergent features are significantly less pronounced when training with `bf16`.

**Broader Impact** Our work serves as a useful counter-example to scholarship which has advanced the notion that certain properties depend only on model scale (Wei et al., 2022a). Rather, our results support the conclusion that optimization choices play a large role in whether emergent properties are present. We believe there is more work to be done here. We also hope that the insights gained from our work illustrate the significant impact the underlying hardware can have on PTQ. Currently, `bf16` training is possible on TPUs and only very recently introduced to A100 & H100 GPUs. Finally, we belive our results present an impactful formula for training models which are inherently easier to quantize at scale, making these models more accessible for deploying in a variety of deployment environments.

**Limitations and Future Work** We do not vary the architectural design (except for the positional embedding type) and training objective in our experiments given our goal of a controlled experimental set-up and the large computational cost of each variant. We leave this to the future work. In addition, it is worthwhile to explore the impact of finetuning with the recommended optimization choices on make a pre-trained model more quantization-friendly, as well as the theory behind why these optimization choices correlate with better compression results.

# 8  Acknowledgements

We thank João Araújo, Milad Alizadeh and other colleagues in Cohere & Cohere For AI for help-fulfeedback and support. Additionally, we thank Jimmy Ba for helpful discussions surrounding the topic of this paper. We also thank Tim Dettmers for assisting in replicating the outlier dimension definition and results in `int8.LLM()`.

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
