# Appendix

## A   Extended Results & Architecture Details

### A.1   Model Details

| Model Size | Layers | Heads | Hidden Dim. | Max. Learning Rate | Batch Size | $(\beta_1, \beta_2)$ |
|---|---|---|---|---|---|---|
| 410M | 24 | 16 | 1024 | $1.2 \times 10^{-4}$ | 256 | (0.9,0.95) |
| 6B | 28 | 32 | 4096 | $1.2 \times 10^{-4}$ | 256 | (0.9,0.95) |
| 13B | 40 | 40 | 5120 | $1.2 \times 10^{-4}$ | 256 | (0.9,0.95) |
| 52B | 52 | 64 | 8192 | $1.2 \times 10^{-4}$ | 256 | (0.9,0.95) |

*Optimal pre-training hyperparameters for PTQ based on results in Section 4*

| Weight decay | Gradient clipping | Dropout | Half-precision data type |
|---|---|---|---|
| 0.1 | 1.0 | 0 | `bf16` |

Table 2: Model architecture and training details

### A.2   Evaluation Suite

Below is a detailed breakdown of the evaluation suite we evaluate our models with.

| Benchmark | Task Type | Evaluation Metric |
|---|---|---|
| Copa (test set)(Wang et al., 2019) | MC Completion | MC Accuracy |
| Copa100 (dev set) (Wang et al., 2019) | MC Completion | MC Accuracy |
| HellaSwag (Zellers et al., 2019) | MC Completion | MC Accuracy |
| PIQAValidation (Bisk et al., 2020) | MC Completion | MC Accuracy |
| StoryCloze (Mostafazadeh et al., 2016) | MC Completion | MC Accuracy |
| WinoGrande (Sakaguchi et al., 2019) | MC Co-referencing | MC Accuracy |
| Paralex (Fader et al., 2013) | Generation | Likelihood (bytes) |
| LAMBADA (Paperno et al., 2016) | Generation | Exact String Match Accuracy |

Table 3: An Overview of the 8 tasks we benchmark the zero-shot downstream performance of trained models. QA and MC denotes Question Answering, and Multiple-choice respectively.

### A.3   Task Result Breakdown

| Model Size | Data type | PIQA | HellaSwag | WinoGrande | LAMBADA | Copa | Copa100 | StoryCloze | Paralex | Average |
|---|---|---|---|---|---|---|---|---|---|---|
| | FP16 | 83.19 | 82.48 | 70.01 | 75.47 | 79.40 | 81.00 | 85.87 | 61.05 | 77.31 |
| 52B | W8A8 | 83.20 | 82.40 | 70.00 | 75.50 | 79.40 | 82.00 | 85.50 | 61.10 | **77.39** |
| | W4 | 80.20 | 72.22 | 66.30 | 66.85 | 78.20 | 83.00 | 82.56 | 60.43 | 73.72 |
| | FP16 | 79.54 | 75.26 | 62.27 | 70.81 | 76.00 | 76.00 | 82.11 | 60.68 | 72.83 |
| 13B | W8A8 | 79.20 | 74.60 | 62.90 | 69.90 | 76.00 | 75.00 | 82.20 | 60.90 | **72.59** |
| | W4 | 76.66 | 60.59 | 57.30 | 46.05 | 73.60 | 76.00 | 77.40 | 59.74 | 65.92 |
| | FP16 | 79.50 | 74.20 | 61.20 | 70.50 | 75.40 | 79.00 | 81.50 | 60.10 | 72.67 |
| 6B | W8A8 | 79.50 | 73.70 | 61.40 | 70.00 | 74.60 | 77.00 | 81.00 | 60.20 | **72.18** |
| | W4 | 76.93 | 62.92 | 56.43 | 55.40 | 74.00 | 72.00 | 77.21 | 59.01 | 66.74 |
| | FP16 | 70.40 | 46.90 | 50.80 | 48.80 | 65.40 | 65.00 | 70.50 | 57.10 | 59.36 |
| 410M | W8A8 | 70.00 | 46.80 | 51.50 | 47.80 | 64.00 | 64.00 | 69.70 | 57.00 | **58.85** |
| | W4 | 67.19 | 43.08 | 50.59 | 37.71 | 62.80 | 64.00 | 67.47 | 54.60 | 55.93 |

Table 4: Our fully trained models with hyper-parameters outlined in Table 2 show minimal PTQ degradation.

| | Data type | PIQA | HellaSwag | WinoGrande | LAMBADA | Copa | Copa100 | StoryCloze | Paralex | Average | Average % Diff |
|---|---|---|---|---|---|---|---|---|---|---|---|
| wd=0.1 (gc=none) | FP16 | 75.35 | 60.58 | 55.41 | 56.59 | 73.20 | 71.00 | 77.02 | 58.75 | 65.99 | -0.09 |
| | INT8 | 75.35 | 60.36 | 55.25 | 57.69 | 73.20 | 70.00 | 76.70 | 58.62 | 65.90 | |
| wd=0.01 (gc=none) | FP16 | 75.03 | 60.55 | 55.25 | 59.01 | 72.00 | 69.00 | 77.53 | 58.99 | 65.92 | -0.26 |
| | INT8 | 74.48 | 60.10 | 55.49 | 60.14 | 72.40 | 67.00 | 77.21 | 58.87 | 65.71 | |
| wd=0.001 (gc=none) | FP16 | 75.90 | 60.71 | 55.80 | 58.16 | 73.00 | 71.00 | 76.64 | 58.50 | 66.21 | -1.36 |
| | INT8 | 75.63 | 60.52 | 54.78 | 55.15 | 71.80 | 70.00 | 77.21 | 57.96 | 65.38 | |
| dtype=bf16 (wd=0.1) | FP16 | 75.68 | 60.92 | 55.96 | 57.60 | 71.40 | 71.00 | 75.81 | 59.05 | 65.93 | 0.32 |
| | INT8 | 75.95 | 60.70 | 56.83 | 58.32 | 71.40 | 71.00 | 75.62 | 59.06 | 66.11 | |
| dtype=fp16 (wd=0.1) | FP16 | 73.83 | 59.96 | 55.96 | 56.14 | 72.00 | 67.00 | 75.94 | 58.33 | 64.89 | -0.76 |
| | INT8 | 74.16 | 59.75 | 55.64 | 56.05 | 71.60 | 64.00 | 75.88 | 58.12 | 64.40 | |
| dtype=bf16 (wd=0.01) | FP16 | 74.97 | 60.79 | 55.49 | 57.52 | 72.00 | 68.00 | 75.88 | 58.74 | 65.42 | 0.77 |
| | INT8 | 75.08 | 60.51 | 55.88 | 59.31 | 72.60 | 69.00 | 76.00 | 58.84 | 65.90 | |
| dtype=fp16 (wd=0.01) | FP16 | 74.81 | 58.11 | 54.93 | 57.31 | 70.20 | 71.00 | 74.67 | 58.25 | 64.91 | -1.73 |
| | INT8 | 73.61 | 56.90 | 54.22 | 53.02 | 71.60 | 69.00 | 74.67 | 57.92 | 63.87 | |
| gc=1.0 (wd=0.001) | FP16 | 74.65 | 60.03 | 54.78 | 59.01 | 71.60 | 67.00 | 76.96 | 58.92 | 65.37 | 0.41 |
| | INT8 | 74.92 | 59.91 | 54.62 | 59.69 | 72.20 | 68.00 | 76.96 | 58.90 | 65.65 | |
| gc=none (wd=0.001) | FP16 | 75.90 | 60.71 | 55.80 | 58.16 | 73.00 | 71.00 | 76.64 | 58.50 | 66.21 | -1.36 |
| | INT8 | 75.63 | 60.52 | 54.78 | 55.15 | 71.80 | 70.00 | 77.21 | 57.96 | 65.38 | |
| dropout=0.0 | FP16 | 75.68 | 60.92 | 55.96 | 57.60 | 71.40 | 71.00 | 75.81 | 59.05 | 65.93 | 0.32 |
| | INT8 | 75.95 | 60.70 | 56.83 | 58.32 | 71.40 | 71.00 | 75.62 | 59.06 | 66.11 | |
| dropout=0.1 | FP16 | 74.76 | 58.87 | 54.38 | 57.23 | 71.60 | 68.00 | 76.45 | 58.36 | 64.96 | 0.31 |
| | INT8 | 74.27 | 58.70 | 54.85 | 58.35 | 71.80 | 68.00 | 76.96 | 58.18 | 65.14 | |
| dropout=0.4 | FP16 | 74.92 | 55.80 | 54.70 | 58.98 | 71.00 | 66.00 | 74.03 | 57.91 | 64.17 | -0.27 |
| | INT8 | 74.76 | 55.77 | 54.14 | 59.69 | 69.40 | 66.00 | 74.09 | 57.95 | 63.98 | |
| dropout=0.8 | FP16 | 67.79 | 30.87 | 50.51 | 37.12 | 67.00 | 65.00 | 61.94 | 29.16 | 51.17 | -0.57 |
| | INT8 | 67.79 | 30.75 | 50.12 | 36.52 | 66.60 | 65.00 | 61.74 | 28.91 | 50.93 | |

# B Extended Weight & Activation Analysis

## B.1 Weight Distributions

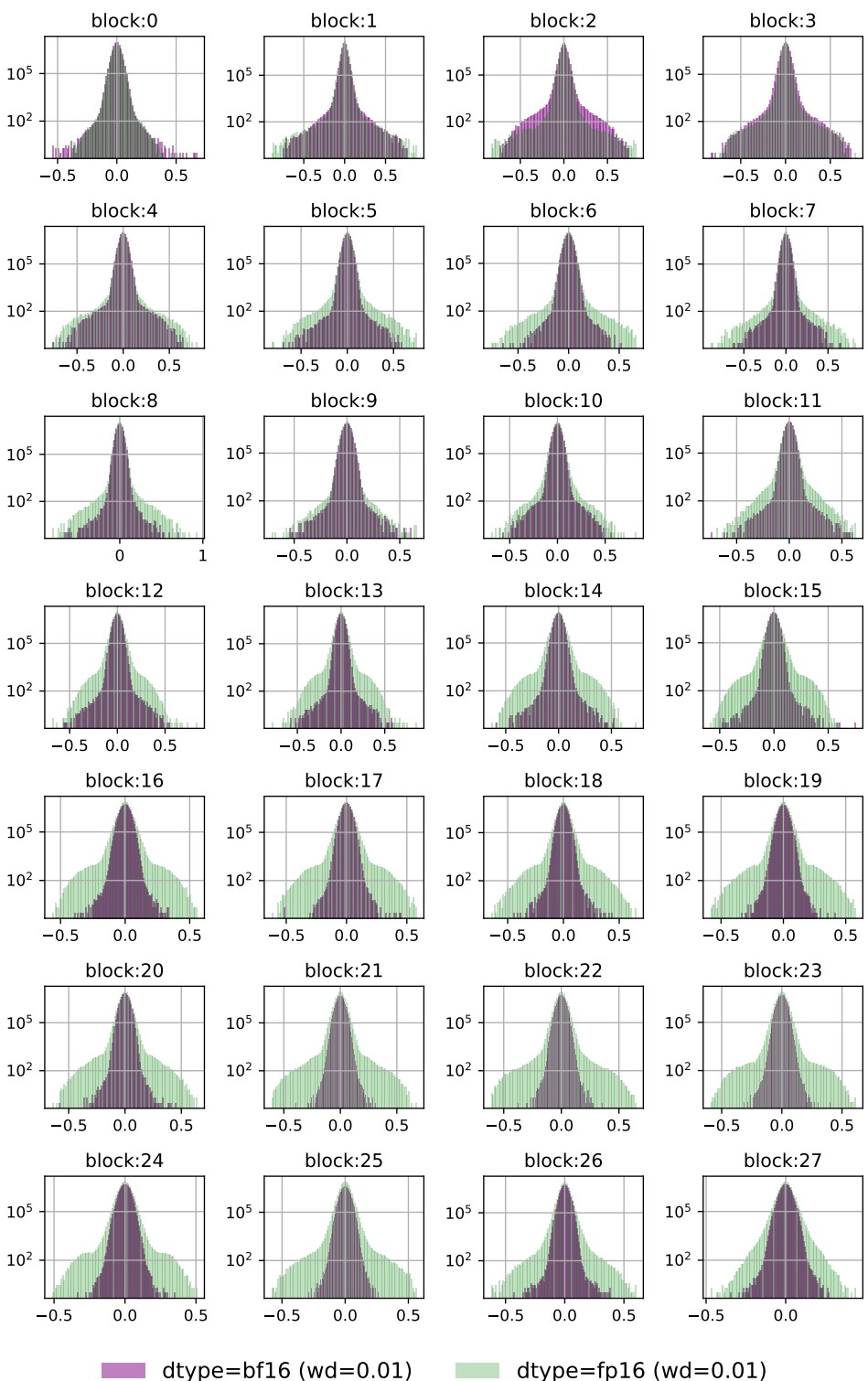

Figure 6: Weight distributions for `attn-kqv-proj` layers comparing `fp16` and `bf16` variants.

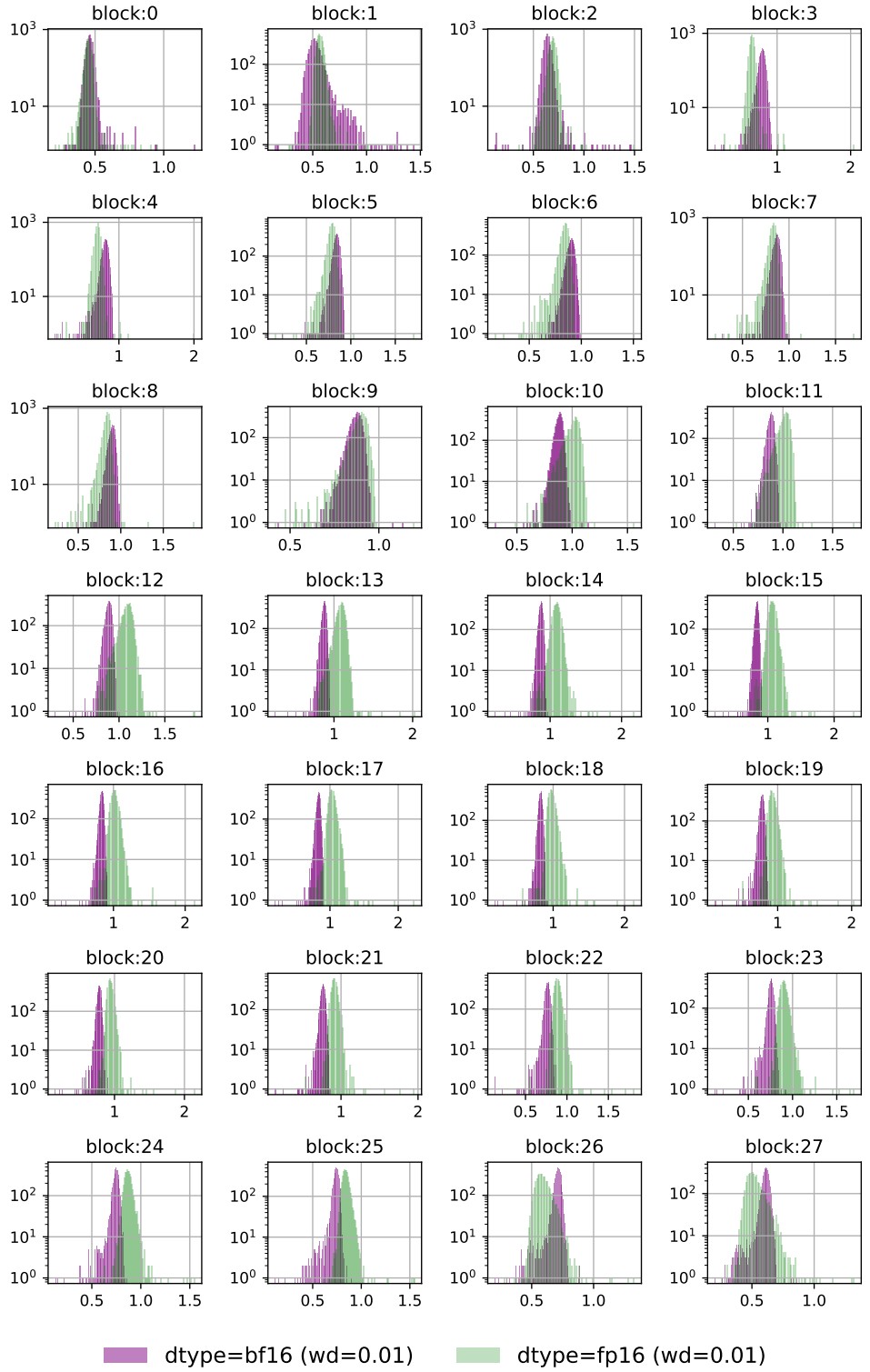

Figure 7: Gain parameter distributions of the first layernorm comparing `fp16` and `bf16` variants.

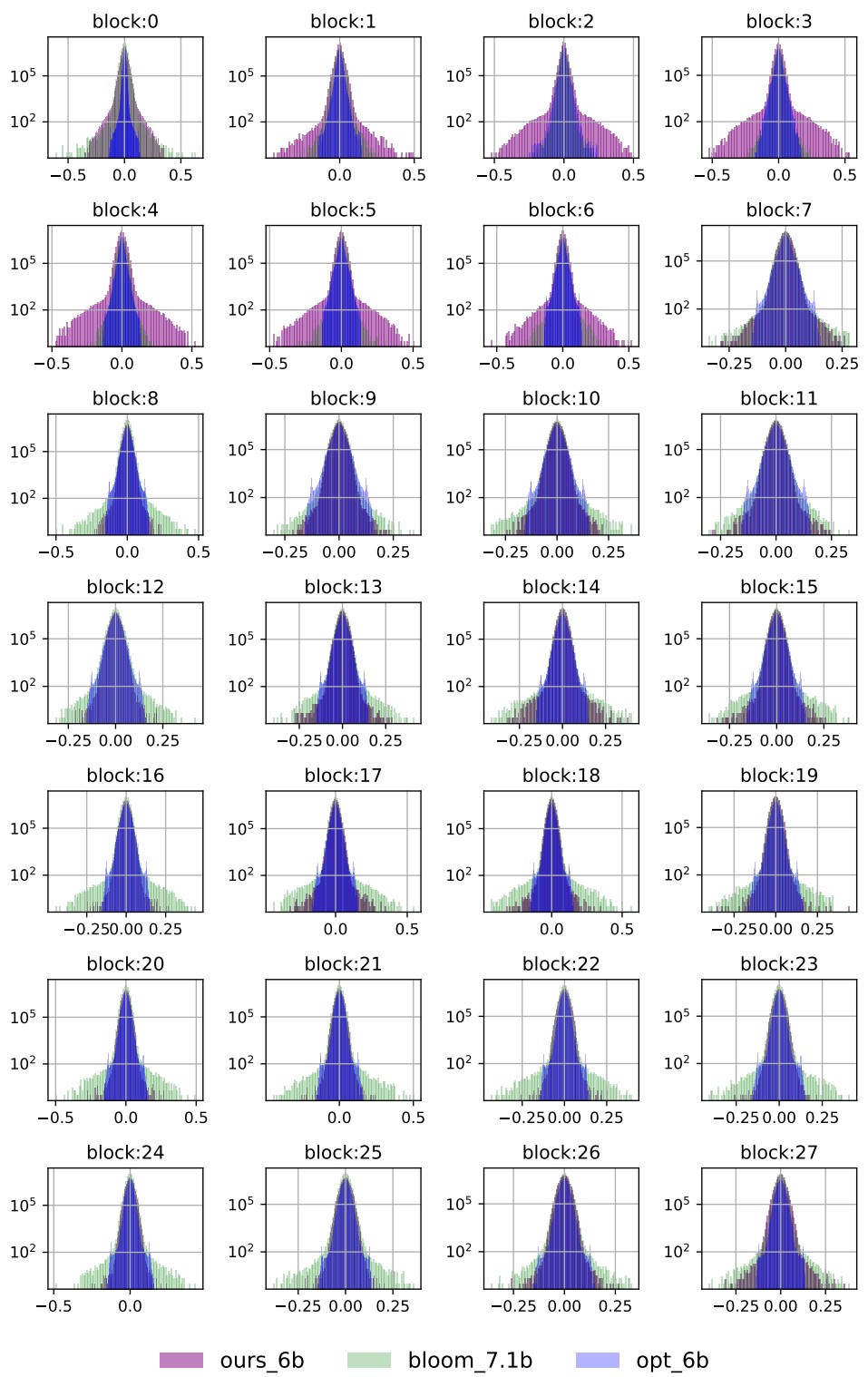

Figure 8: Weight distributions of `attn-kqv-proj` layers comparing our model against OPT-6B & BLOOM-7.1B

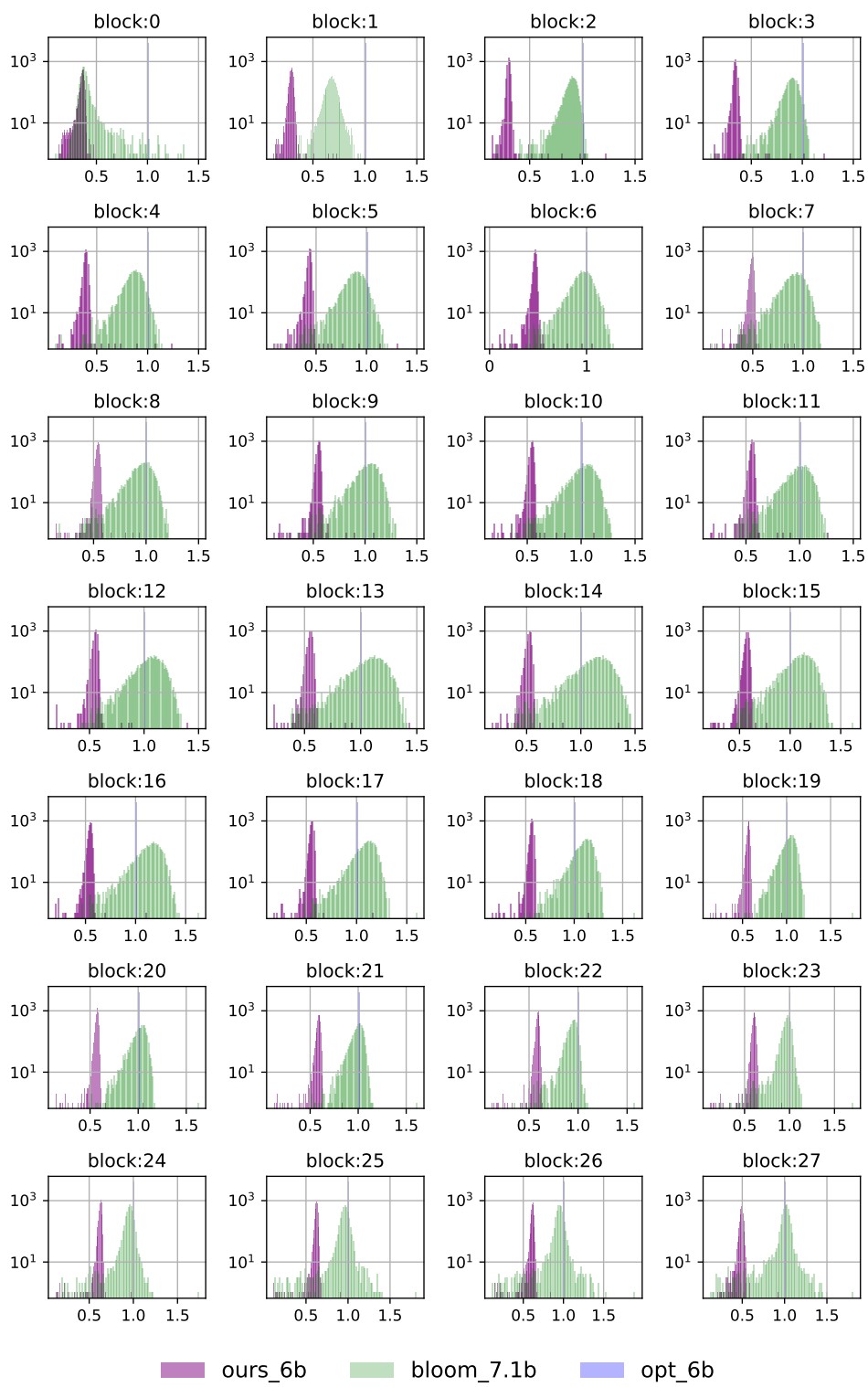

Figure 9: Gain parameter distributions of the first layernorm comparing our model against OPT-6B. Note that in OPT-6B, all layernorm gain parameters across the network are set to 1.0.

## B.2 All Layers Analysis

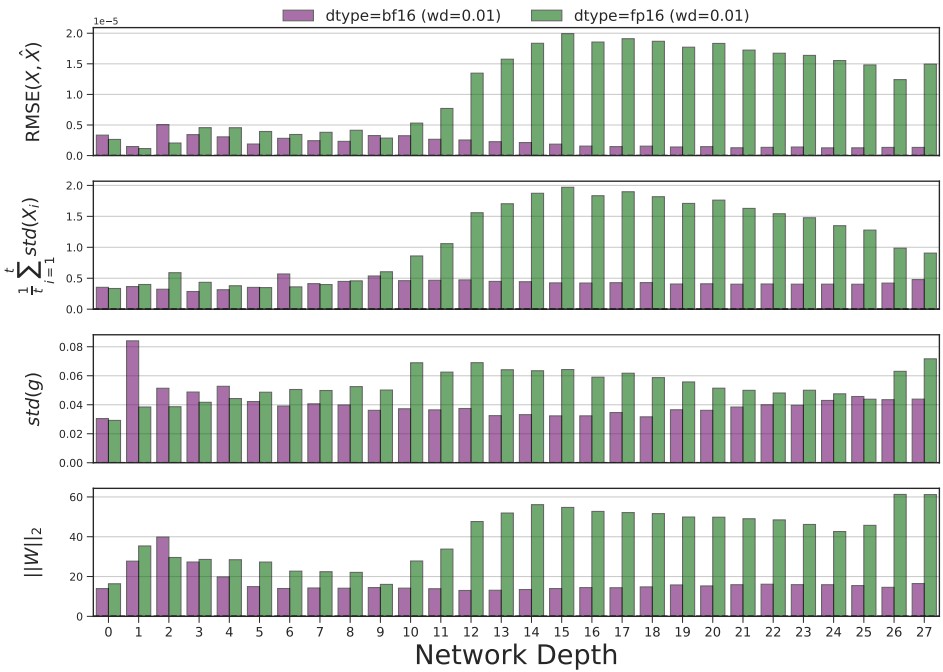

Figure 10: Input activation RMSE and token STD are higher in the `fp16` variant for most `attn-kqv-proj`. A similar trend exists for the first layernorm gain STD and weight spectral norm.

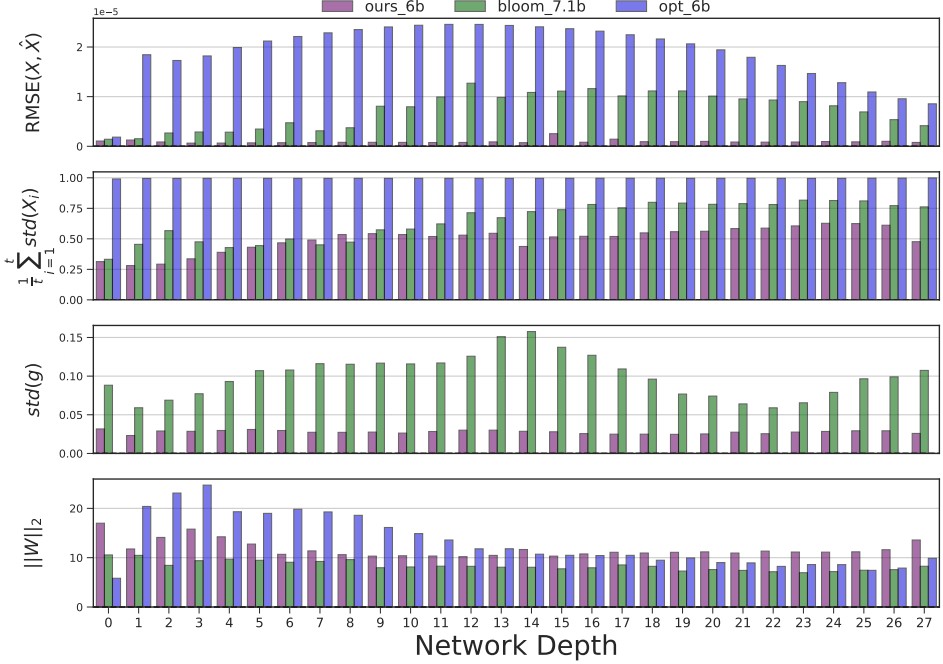

Figure 11: In line with Figure 10, the less quantization robust OPT-6B model has significantly higher RMSE($\mathbf{X},\hat{\mathbf{X}}$) and average token activation STD for all `attn-kqv-proj` layers. Note that all gain parameters seem to have been hard coded to 1.0 in OPT-6B.

### B.3 Outlier analysis

We classify a feature dimension as an outlier dimension if the *same* dimension is classified as an outlier across 20 random samples from the C4 validation set (Raffel et al., 2020). These samples are fixed when experimenting with different outlier definitions. As mentioned in Section 4, we experimented with the definition outlined in Dettmers et al. (2022) where a hidden feature dimension is classified as outlier dimension if the activation magnitudes (denoted as $\alpha$) are greater than 6.0 ($\alpha > 6$) at more than 25% of layers and 6% of tokens. However, we find *0* outlier dimensions across all of our 6B variants and fully trained models using this definition. As a result, following the methodology outlined in Dettmers et al. (2022), we manually searched for the lowest threshold such that only one dimension is classified as outlier in our smallest (410M) fully trained model. As shown in Table 5, even classifying outlier dimensions using the searched threshold of 4.2 resulted in most variants having *0* outlier dimensions. We further experimented with not fixing the threshold and using *z-score* outlier detection i.e. classify a feature as a high-magnitude if $\alpha > C\sigma + \mu$, where $\sigma$ and $\mu$ denote sample standard deviation and mean. We were not able to establish clear trends with this method either as shown in Table 5.

Table 5: Outlier statistic using different thresholding rules **Top Table:** constant threshold $\alpha > 4.2$ **Bottom Table:** adaptive threshold $\alpha > 4\sigma_{token} + \mu_{token}$

| Variant | #Outliers | %Seq Affected | %Layers Affected |
|---|---|---|---|
| `{all other variants}` | 0 | 0 | 0 |
| `dtype=fp16 (wd=0.01)` | 6 | 68.4 | 65.5 |
| `ours_410M` | 1 | 25.0 | 26.3 |
| `ours_6B` | 0 | 0 | 0 |
| | | | |
| `dropout=0.1` | 2 | 44.0 | 40.5 |
| `dropout=0.4` | 2 | 44.9 | 42.9 |
| `dropout=0.8` | 1 | 19.2 | 25 |
| `wd=0.1 (gc=none)` | 2 | 28.0 | 39.3 |
| `wd=0.01 (gc=none)` | 0 | 0 | 0 |
| `wd=0.001 (gc=none)` | 2 | 34.3 | 34.5 |
| `dtype=bf16 (wd=0.1)` | 2 | 33.0 | 36.9 |
| `dtype=fp16 (wd=0.1)` | 2 | 41.1 | 42.9 |
| `dtype=bf16 (wd=0.01)` | 0 | 0 | 0 |
| `dtype=fp16 (wd=0.01)` | 8 | 66.2 | 64.3 |
| `ours_410M` | 55 | 88.2 | 86.2 |
| `ours_6B` | 7 | 65.5 | 56 |

## C Extended Literature Review

The need for compression techniques that scale to large language model settings has become increasingly urgent with larger and larger models (Treviso et al., 2022; Yao et al., 2023). There has been a renewed focus on efficiency techniques (Gale et al., 2019; Ogueji et al., 2022; Ahia et al., 2021). Quantization as a form of model compression of large language models has become increasingly relevant as a way to minimize memory requirements and minimize compute intensity (Dettmers et al., 2022; Xiao et al., 2022; Frantar et al., 2022; Park et al., 2022a; Kim et al.).

**Model Efficiency at Inference Time** Research in model compression mostly falls in the categories of quantization techniques (Jacob et al., 2018; Courbariaux et al., 2014; Hubara et al., 2016; Gupta et al., 2015), efforts to start with a network that is more compact with fewer parameters, layers or computations (architecture design) (Howard et al., 2017; Iandola et al., 2016; Kumar et al., 2017), student networks with fewer parameters that learn from a larger teacher model (model distillation) (Hinton et al., 2015) and finally pruning by setting a subset of weights or filters to zero (Louizos et al., 2017; Wen et al., 2016; LeCun et al., 1990; Hassibi et al., 1993a; Ström, 1997; Hassibi et al., 1993b; See et al., 2016; Narang et al., 2017; Frantar & Alistarh, 2023; Sanh et al., 2020). Often, a

combination of compression methods might be applied. For example, pruning might be combined with other efficiency-improving methods, e.g. quantization or faster search algorithms.

**Quantization Techniques** Quantization can be used to speed up inference and relax hardware requirements, as has been shown for e.g., 8-bit (Quinn & Ballesteros, 2018), 4-bit (Aji & Heafield, 2020) and recently also below 3-bit quantization (Park et al., 2022a) of neural machine translation models. Park et al. (2022a) utilize non-uniform quantization to achieve high compression ratios. However, this method requires the use of specialized kernels for compressed (2-bit/4-bit) weights and floating point activations and involve finding the binary representations using expensive iterative search or QAT. Similar to Park et al. (2022a), Frantar et al. (2022) also demonstrate compression of model parameters to 3 or 4-bit precision allowing inference off a single A100 GPU. However, they also perform weight-only quantization limiting the speedup as the activations are kept at higher precision (FP16).

Quantization reduces the number of bits needed to represent model weights which minimizes both the memory and latency required to serve a model. Often, the goal is to quantize the bit representation while preserving equivalent performance. Quantization approaches can be broadly categorized into:

1. **Quantization-aware training** (QAT) (Zafrir et al., 2019; Krishnamoorthi, 2018) involves pre-training with simulated quantization, enabling parameters to adjust to lower precision grids. This requires estimating the derivative of non-differentiable quantization operators, performing full backpropagation throughout the entire model, and training with the entire training dataset. However, this method can be computationally expensive, particularly for large language models.

2. **Quantization-aware finetuning** (QAF) (Yao et al., 2022; Frantar et al., 2022; Zhuo et al., 2022; Li et al., 2021; Hubara et al., 2020; Nagel et al., 2020) is a more efficient approach that utilizes a pretrained model and a small subset of training data (i.e., hundreds of samples) to optimize performance under quantization. By simulating quantization and optimizing a small range of parameters at a time, no backpropagation is needed while the quantization loss can be reduced.

3. **One-shot post-training quantization** (PTQ) (Xiao et al., 2022; Dettmers et al., 2022) unlike QAT and QAF, does not involve optimization. Instead, it directly maps data from a high precision range to a low precision range based on a hand-picked mapping function.

Given the complexities of successfully training a large language model (Zhang et al., 2022; Rae et al., 2021), post-training quantization (PTQ) methods are extremely attractive as these techniques require the least modification to pretrained parameters. This is the focus of our exploration in this work.

### C.1 Introduction to Post-Training Integer Quantization Approaches

Below section introduces widely used quantization methods and provides context about the differences between these methods. The quantization strategy for weights and activations can be broadly classified into three categories:

### C.1.1 Weight-only Quantization

Weight-only quantization has proven extremely effective in making large language models accessible by enabling inference in a resource-constrained environment while maintaining the FP16 model quality (Gerganov, 2023; Frantar et al., 2022; Zeng et al., 2022; Sheng et al., 2023). Weight-only quantization provides improvements in latency due to a reduction in time taken for parameter fetching from GPU global memory, however, the actual Matrix-Matrix multiplication (GEMM) operations are carried out at higher precision in FP16 - allowing modest gains on platforms without dedicated lower-precision GEMM operations support.

### C.1.2 Weight and Activation Quantization

As large language models are scaled, progressively they become compute-bound and the improvements due to weight-only quantization stagnate. However, in this regime, using efficient kernels that leverage specialized lower-precision cores in modern GPUs to directly perform the actual Matrix-Matrix multiplication operation at lower precision enables large latency gains - due to the increased

throughput of INT8 tensor cores over FP16 Tensor Cores (Nvidia). As this quantization technique scales the best, this is our focus in this work.

To-date quantization of *both* the activations and weights of very large models (>6.7B parameters) has proven challenging - leading to a large drop in performance (Dettmers et al., 2022).

### C.1.3   Quantization by Mixed-Precision Decomposition

In the quantization strategies mentioned above, even though the various weights and activations might be stored in different precisions; all the computations in a single operation are carried out at the same precision (FP16 or INT8). In contrast, `LLM.int8()` (Dettmers et al., 2022) proposes to decompose the matrix multiplication to compute a small fraction of elements at a higher precision (FP16) while the bulk of the computations is performed at low precision (INT8). This approach has a similar footprint to that of weight-only quantization but practical latency gains are limited or potentially worse. While this approach has theoretical latency benefits due to the bulk of the computation being performed at lower precision, in practice without specialized hardware (Dash et al., 2022; Dash & Mukhopadhyay, 2020), the lack of specialized kernels on GPUs and additional kernel calls required to ready the inputs and weights for mixed-precision computation negates the projected benefits. In this work we focus on exploring optimization choices which mitigate quantization trade-offs for both *weight-only quantization* and the far more challenging *weight and activation quantization*.

We benchmark end-to-end inference speed for our weight and activation quantization recipe on our 52B model for both compute-bound (large batch size, long input sequence and short generation) and memory-bandwidth-bound (small batch size, short input sequence and long generation) use cases (Table 6). We use standard CUDA kernels that enable INT8 matrix multiplication. Across all settings, we find that our vector-wise PTQ approach provides a 1.4-1.5x latency speedup (or 1.4-1.5x throughput improvement) and around 40% memory footprint reduction over half-precision inference. We note that mixed-precision decomposition (LLM.int8()) is slightly slower than half-precision inference (Dettmers et al., 2022). Therefore, our PTQ approach is significantly faster than LLM.int8(). Additionally, Dettmers et. al. also show that LLM.int8() leads to slower matrix multiplication compared to vector-wise PTQ.

Table 6: Latency, throughput and memory savings of INT8 Vector-wise quantization in various input length, batch size, and output length settings for our 52B parameter model ran on 4 A100 40GB GPUs. Benchmarks are conducted with standard CUDA kernels that enable INT8 matrix multiplication.

| | Batch Size | Input Length | Output Length | Half-Precision (fp16) baseline | INT8 Vector-wise | Gain |
|---|---|---|---|---|---|---|
| **Latency (ms)** | | | | | | |
| Memory-bandwidth bound | 1 | 60 | 1024 | 31015 | 22151 | 1.40 |
| | 2 | 60 | 1024 | 31403 | 22311 | 1.41 |
| | 4 | 60 | 1024 | 31789 | 22668 | 1.40 |
| Compute bound | 8 | 1024 | 1 | 1116 | 766 | 1.46 |
| | 16 | 1024 | 1 | 2224 | 1561 | 1.43 |
| | 8 | 2048 | 1 | 2254 | 1591 | 1.42 |
| **Throughput (tokens generated per second)** | | | | | | |
| Memory-bandwidth bound | 1 | 60 | 1024 | 33.0 | 46.2 | 1.40 |
| | 2 | 60 | 1024 | 32.6 | 45.9 | 1.41 |
| | 4 | 60 | 1024 | 32.2 | 45.2 | 1.40 |
| Compute bound | 8 | 1024 | 1 | 0.9 | 1.3 | 1.44 |
| | 16 | 1024 | 1 | 0.5 | 0.6 | 1.42 |
| | 8 | 2048 | 1 | 0.4 | 0.6 | 1.43 |
| **GPU Peak Memory (GB)** | | | | | | Reduction |
| Memory-bandwidth bound | 1 | 60 | 1024 | 29.1 | 17.0 | 42% |
| | 2 | 60 | 1024 | 29.7 | 17.6 | 41% |
| | 4 | 60 | 1024 | 30.8 | 18.7 | 39% |
| Compute bound | 8 | 1024 | 1 | 32.8 | 20.6 | 37% |
| | 16 | 1024 | 1 | 36.8 | 24.6 | 33% |
| | 8 | 2048 | 1 | 36.7 | 24.6 | 33% |