# OpenReview forum: "Intriguing Properties of Quantization at Scale"
_NeurIPS.cc/2023/Conference — NeurIPS 2023 poster_

### Official Review · Reviewer_TcCw · 2023-07-04

**Soundness:** 2 fair
**Presentation:** 2 fair
**Contribution:** 3 good
**Rating:** 5
**Confidence:** 3

**Summary:**



The paper empirically studies the effect of optimization/regularization hyperparameters -- weight-decay, dropout, gradient-clipping -- on post-training quantization (PTQ) performance of large language models (LLMs). Specifically, it performs PTQ to int8 by simple affine scaling, adopts a GPT-style architecture and trains models with various hyperparameter settings, and finds that higher levels of regularization generally helps lower the performance degradation after quantization. The paper finds that at 52B parameters, training with the best hyperparameters results in a 0.26% average performance degradation, and shows that activation outliers are not a universal emergent property of LLMs. Finally it computes various statistics of weights and activations such as std and spectral norm, for the proposed model v.s. other models, arguing they may offer insight about robustness to quantization.



**Strengths:**

1. The paper is mostly clearly written and easy to follow.

2. The problem setting appears well motivated; I'm not an expert in NLP but a quick read of related work strongly suggests to me the difficulty of quantizing transformer-based language models, in particular when both the network weights and activations are quantized.

3. The proposed method for mitigating performance drop due to quantization is surprisingly simple -- just use a higher amount of regularization. The empirical result (0.26% average performance degradation) seems quite good compared to much larger performance drops experienced by other existing models at int8 precision; however this point remains to be clarified (see weakness).

**Weaknesses:**

1. In the abstract, it is claimed "we find that outlier dimensions are not an inherent product of scale, but rather sensitive to the optimization conditions present during pre-training." Where exactly is this result established? The main experiments (section 4) mostly focused on the performance on zero-shot NLP tasks, rather than outlier dimensions. Moreover, "outliers" are frequently mentioned but not defined until later in the paper, which makes this argument hard to follow and it's unclear what definitive role outliers play in post-training quantization performance.

2. The paper acknowledges that there's many possible factors contributing to a LLM's sensitivity to post-training quantization, including model choices (architecture, number of parameters/scale) and optimization/regularization choices, and recent work show conflicting findings between different model families (line 130-132). While paper focuses on the effect of regularization choices using a GPT style architecture (line 178), can we say anything about the effect of different model architectures on quantized performance? e.g., would the proposed methodology apply to an OPT architecture? As is currently written, the paper cannot exclude the possibility that the relatively small performance drop experienced by their trained model is a result of the model architecture, rather than the choice of optimization/regularization.

3. Related to the above: I'm not familiar with LLM evaluation, and it's unclear how to interpret the 0.26% average degradation of the paper's largest (52B) model (line 260). Is it good or bad? It certainly seems impressive compared to the 42% drop encountered by the most comparable OPT model (line 263), but it seems pretty normal compared to performance drops of 0.09% and 1.36% when using various optimization hyperparameters (Figure 2) within the GPT-style model considered by the paper.

**Questions:**

See weaknesses above.

---

> ### Author Rebuttal · Authors · 2023-08-09
>
> We thank **R TcCw** for their valuable feedback and noting that the problem of quantizing both activations and weights is “well motivated “ and that our paper is “clearly written” and “easy to follow”.
>
>
> > While paper focuses on the effect of regularization choices using a GPT style architecture (line 178), can we say anything about the effect of different model architectures on quantized performance? e.g., would the proposed methodology apply to an OPT architecture?
>
> Thank you for the feedback. We clarify that both our model and OPT are decoder-only models and have many architectural similarities. We provide more details on the architectural similarities of these models in the global response. In summary, we believe our results are transferable to the OPT architecture.
>
> > I'm not familiar with LLM evaluation, and it's unclear how to interpret the 0.26% average degradation of the paper's largest (52B) model (line 260). Is it good or bad? It certainly seems impressive compared to the 42% drop encountered by the most comparable OPT model (line 263), but it seems pretty normal compared to performance drops of 0.09% and 1.36% when using various optimization hyperparameters (Figure 2) within the GPT-style model considered by the paper.
>
> We would like to clarify that 0.26% degradation for the 52B model is 1) across the full set of 8 tasks we describe in lines 203-211 2) **after training from random initialization for more than 200k  steps, not after only 75k steps**. With this fully trained 52B model, we achieve 0.0% degradation across HellaSwag, PIQA, WinoGrande, and LAMBADA (all of which are included in the previously mentioned 8 tasks), whereas the similarly sized OPT-66B which has a very similar architecture to ours, shows an extremely sharp drop (42%). Thus, the 0.26% degradation across the full eval suite and 0.0% across the previously mentioned 4 tasks is indeed very impressive.
>
> The PTQ degradations presented in Figure 2 correspond to 6B models which are only trained for 75k steps, and are not directly comparable with that of fully trained models. We use these results to establish the trends in PTQ performance w.r.t. optimization choices. As shown in Figure 3 (right), performance drop, which is due to some optimization choices such as fp16 training, increases with step count. OPT models were also fully trained using fp16, which supports our insight from the 75k 6B checkpoints, that fp16 training leads to notably worse PTQ performance, at scale.
>
>
> >In the abstract, it is claimed "we find that outlier dimensions are not an inherent product of scale, but rather sensitive to the optimization conditions present during pre-training." Where exactly is this result established? The main experiments (section 4) mostly focused on the performance on zero-shot NLP tasks, rather than outlier dimensions. Moreover, "outliers" are frequently mentioned but not defined until later in the paper, which makes this argument hard to follow and it's unclear what definitive role outliers play in post-training quantization performance
>
> We will update the final manuscript to provide a focused discussion on outlier dimensions earlier in the paper, to accompany the downstream PTQ degradations as opposed to Appendix B.3, where it currently resides.
>
> When we use the term outliers, we are referring to large magnitude features in the activations as defined and investigated in LLM.int8(). The authors provide study of activation outliers and provide evidence that they are responsible for performance drops after activation quantization at scales beyond 6.7B. Concretely, since the scaling coefficient used in INT8 quantization to rescale all values to be within INT8 range (-127,127), is directly proportional to the maximum values, large magnitude outliers lead to asymmetric and wide quantization distributions where most quantization bins are empty and small values are quantized to zero, causing significant information loss (Dettmers et al.).
>
> In Appendix B.3, We present a detailed description of our reproduction of outlier dimensions as outlined in LLM.int8(). As the axis with the largest effect on PTQ performance, fp16 training increases the number of outlier dimensions to 8 while bf16 training corresponds to only 0 outliers. Note that the definition of outlier dimensions in LLM.int8(), was hard to generalize due to hard-coded magnitude threshold and number of affected layers and tokens, which failed to adapt to networks with different order of magnitude of activation. Therefore, we explore alternative metrics such as standard deviation (STD) of input token activations that are more easily generalizable. Comparing our 6B model with OPT-6B and BLOOM-7.1B, STD values of the OPT-6B and BLOOM-7.1B are consistently higher than that of our model, up to 3.6x and 2x, respectively that affirms the generalizability of this metric.

---

> > ### Comment · Reviewer_TcCw · 2023-08-15
> >
> > Thank you for the rebuttal, which has addressed some of my concerns. I raised my score to borderline accept, as my remaining issues are more concerned with presentation, and I trust the authors' to address this in the future revision.
> >
> > 1. Thanks for the additional experiment validating the findings on alternative architectures. This gives me more confidence about the validity of the approach.
> >
> > 2. Thanks for the clarification regarding the 42% drop.  Please add this explanation to provide better context for this result in future revisions -- specifically how Dettmers et al.'s setup / training procedure differed from yours that could have led to such a drastic difference. It would also be more convincing to connect these differences to corresponding "bad" hyperparameter settings in your experiments, which also experience bigger accuracy drops. This would essentially corroborate the findings by Dettmers et al.
> >
> > 3. Yes, please discuss early on the definition of "outlier dimension" and why the paper measures it the way it does, and specifically how it relates to std. I still find the claim that "we find that outlier dimensions are not an inherent product of scale, but rather sensitive to the optimization conditions present during pre-training." problematic, as there is no easy way to see this from the experiments. So consider rewriting the statement in terms of the "std/variability" of the parameters (which could indicate quantization error), which the experiments are actually about.
> >
> > Overall, I think the contribution and potential practical impact is valuable enough, but I have issues with the level of rigor and imprecise/overly general claims that can be misleading (see above). The title is also a bit grandiose in my opinion (what are the actual "intriguing properties of quantization at scale"? a more straightforward title about a recipe for mitigating post-training quantization error in LLMs could have served well), as are general claims about showing that "reoccurring activation outliers are not a universal emergent property of LLMs at scale".

---

> > > ### Author Response · Authors · 2023-08-15
> > >
> > > Dear reviewer **TcCw**, thank you for raising your score in light of the clarifications to the manuscript and experiments on the architectural transferability of our method . We are glad that we were able to address your concerns and are grateful for your valuable suggestions which provided us the opportunity to clarify the significance of our results and run additional experiments confirming that our insights on the effect of pertaining optimization choices on PTQ, are applicable to models with architectural differences such as the type of positional embedding.
> > >
> > > We would like to thank the reviewer again for their positive update and detailed feedback on the presentation of the current manuscript. We will update the final manuscript to address these concerns. Please let us know if there are any other concerns/questions we can address during the rebuttal period and we would be happy to clarify.

---

### Official Review · Reviewer_xK34 · 2023-07-07

**Soundness:** 3 good
**Presentation:** 3 good
**Contribution:** 3 good
**Rating:** 7
**Confidence:** 1

**Summary:**

Recent research suggests that the trade-off incurred by quantization is an emergent property, resulting in sharp performance drops in models with over 6 billion parameters. In this study, the authors investigate whether quantization cliffs in performance are solely determined by model scale. Considering the increased research focus on understanding the emergence of certain properties at scale, the authors provide a valuable counter-example.

The authors propose the possibility of optimizing a quantization-friendly training recipe that suppresses outliers in large activation magnitudes. They find that outlier dimensions are not an inherent consequence of scale but rather sensitive to the optimization conditions during pre-training. This discovery opens up avenues for more efficient quantization methods and raises questions about the inherent nature of other emergent properties and their sensitivity to optimization and architecture design choices.

The authors successfully apply their approach to quantize models of various sizes, ranging from 410 million to 52 billion parameters, with minimal performance degradation. This showcases the practical implications of their work and its potential to facilitate effective quantization across a broad range of model scales.

**Strengths:**

The authors conducted a controlled experiment, maintaining consistent architecture while varying key optimization choices. This approach allowed them to systematically investigate the factors influencing sensitivity to quantization.

The study rigorously analyzed the impact of optimization choices on sensitivity to quantization in models ranging from 410 million to 52 billion parameters. By carefully controlling these factors, the authors demonstrated that reoccurring activation outliers are not a universal emergent property of large language models at scale.

Despite the massive scale of 52 billion parameters, the authors achieved minimal performance degradation (0.26%) across eight tasks when applying quantization to both activations and weights. This showcases the effectiveness of their optimization approach in mitigating the impact of quantization on model performance.

The study provides a detailed analysis of activations and weights, offering insights into the key characteristics influencing sensitivity to quantization. These findings can guide future model design and pre-training strategies to enhance robustness and performance.

**Weaknesses:**

Even though the authors do a thorough empirical analysis, their work doesn't begin to explain why we observer this quantization phenomenon.

**Questions:**

None

**Limitations:**

See weaknesses.

---

> ### Author Rebuttal · Authors · 2023-08-09
>
> We thank **R xK34** for the positive feedback, including their observations that 1) We conducted a series of “controlled experiments, maintaining consistent architecture while varying key optimization choices” which allowed us “systematically investigate the factors influencing sensitivity quantization sensitivity” and infer a set of  PTQ optimal hyperparameters. 2) By using these choice to fully train a 52B parameter model, despite its “massive scale”,  we achieved “minimal performance degradation (0.26%) across eight tasks when applying quantization to both activations and weights”. Which showcases the effectiveness of our “optimization approach in mitigating the impact of quantization on model performance”.
>
> >Even though the authors do a thorough empirical analysis, their work doesn't begin to explain why we observe this quantization phenomenon.
>
> We thank **R xK34** for the opportunity to address this feedback. As they mention, we systematically demonstrate the effect of the key optimization choices on the quantization sensitivity in models ranging from 410M to 52B. We find that high weight-decay, gradient-clipping, no dropout, and most significantly, bf16 as the computation data-type, mitigate unfavorable outlier distributions and lead to the best PTQ performances. Validating these findings, our fully trained 52B model achieves minimal PTQ degradation (0.26%) across 8 tasks.
>
> We relate this quantization robustness to our models’ weight and activation distributions. To show that our recipe leads to a more quantization-friendly activation distribution, we compare RMSE($X,\widehat{X}$) and input token activation STD of our model with that of OPT and BLOOM models for attention projection layers. As shown in Figure 11 (Appendix B.2). RMSE($X,\widehat{X}$) values for OPT-6B and BLOOM-7.1B are up to **33.2x** and **13x** the corresponding values for our fully trained 6B model, respectively. Similarly, STD values are consistently higher for OPT-6B and BLOOM-7.1B than that of our model, up to 3.6x and 2x, respectively.
>
> To explain why some optimization choices generate activation distributions better suited for PTQ, we investigate the models’ weights. Specifically, we find that as the choice with highest impact on PTQ performance of the 6B model variants, bf16 training leads to significantly different model parametrization. Notably, Figure 6 (Appendix B.1) shows that for all attention projection layers (attn-proj-kqv) the model trained with bf16 consistently has a narrower weight distribution compared to the fp16 counterpart, for all layers across the network. Furthermore, bf16 training results in up to 2x higher STD for the gain parameters of the layer norms. (Figure 10, Appendix B.2).
>
> However, we would like to note that **R xK34** raises an important question, and understanding the exact mechanism of how these choices interact with one another can be a new research. We leave this to the future work.

---

### Official Review · Reviewer_vYyu · 2023-07-08

**Soundness:** 3 good
**Presentation:** 4 excellent
**Contribution:** 3 good
**Rating:** 6
**Confidence:** 4

**Summary:**

There is a quantization tradeoff in LLMs, where quantization methods drop in particular for some larger models. This problem has been attributed to large activation magnitude "outliers". The authors note that activation outliers are not an emergent property of LLMs and that instead a byproduct of how the networks were trained.  The authors compare different training strategies  (w/o weight-decay, grad clipping, dropout, half precision) and show that while many configurations can lead to the same performance, after one-shot post training quantization (OPT), the performance can be very different. The authors then train multiple models and show that their results hold for models even at 100B parameters.

**Strengths:**

The paper is clear and easy to follow.  It describes the motivation/problem, related work and the contributions clearly.

**Weaknesses:**

The paper proposes a training strategy for improved quantization.
- As noted by the authors, other work has  also noted that quantization techniques are more stable if the network is trained initialized so that large feature magnitudes are discouraged [1]. Given that this recipe is provided given one architecture I was wondering if the authors could comment on whether they believe the hyperparam configurations will hold across quantization levels/architectures/ pre-training / tasks. Having a paragraph that summarizes this would make the paper more impactful.
- One component of the recipe is using bf16. Fig 5 and other work [2] shows that bf16 is better than fp16, Given that OPT is trained used using fp16, I was wondering how much of the recipe is a benefit of using bf16?
- Could the authors provide a summary of the ablations of the recipe?

[1] Worstman et al. 2023
[2] https://cloud.google.com/blog/products/ai-machine-learning/bfloat16-the-secret-to-high-performance-on-cloud-tpus


**Questions:**

- The authors focus on network retraining for improved OPT quantization. For finetuning many approaches have been developed to improve the weight distribution (such as temperature scaling) Do the authors have an idea of how helpful that would for quantization if applied during finetuning?

---

> ### Author Rebuttal · Authors · 2023-08-09
>
> We thank **R vYyu** for their helpful feedback and their observations noting that our paper is “easy to follow” and “clear” in describing the motivation, problem, and related work.
>
> > As noted by the authors, other work has also noted that quantization techniques are more stable if the network is trained initialized so that large feature magnitudes are discouraged [1]. Given that this recipe is provided given one architecture I was wondering if the authors could comment on whether they believe the hyperparam configurations will hold across quantization levels/architectures/ pre-training / tasks. Having a paragraph that summarizes this would make the paper more impactful.
>
> Thanks for the detailed suggestion, and the opportunity to speak to the transferability of our results. We have provided further details on our position regarding architectural transferability in the global response, but in short, we believe slight architectural differences such as the type of positional embedding do not impede the transferability of our results.
>
> We also benchmarked on diverse types of tasks in our evaluation suite (lines 203-207), covering 8 tasks of 4 different types (Multiple choice (MC) completion, MC Co-referencing, Generation, and Question Answering). Hence, we believe that our results are not dependent on the nature of the downstream evaluation task. We will add a paragraph clarifying these details to summarize our position on the transferability of our results to settings different than ours to the final manuscript.
>
> >One component of the recipe is using bf16. Fig 5 and other work [2] shows that bf16 is better than fp16. Given that OPT is trained used using fp16, I was wondering how much of the recipe is a benefit of using bf16?
>
> We believe that bf16 training has a significant impact on downstream PTQ as fp16 training leads to the largest swing in degradation amongst all experimental axes (Figure 3, Left). When coupled with weight decay of 0.01 and otherwise PTQ optimal hyperparameters, fp16 training leads to high degradation as early as 15k steps and at 75k steps, leads to the largest drop in PTQ performance out of all the early checkpoint model variants (Figure 3, Right). Simply switching to bf16 training leads to a 6.7x reduction in PTQ degradation (inferred from Figure 2a, second column) which is the largest for any hyperparameter we studied. Moreover, as we briefly mention in lines 244-247, BLOOM-176B which is trained with bf16, is significantly more robust to PTQ than OPT-175B which is trained with fp16. This claim is based on the results published in SmoothQuant (Table 4) where INT8 quantization of both activations and weights leads to a 4% degradation for BLOOM-176B where OPT-175B shows a 39.2% drop.
>
> >Could the authors provide a summary of the ablations of the recipe?
>
> To understand the effects of dropout, gradient-clipping, weight-decay, and training computation datatype, **per experimental axis we train models from random initialization, varying each hyper-parameter while keeping others the same** to establish trends by comparing the mean PTQ degradation across 8 diverse tasks. In total, we train 13 6B parameter models from random initialization for 75k steps each and draw the following insights: we find that high weight-decay, gradient-clipping, no dropout, and most significantly, bf16 as the computation data-type, lead to the best PTQ performances in their respective axis. To validate that these insights carry to different scales and full training, we fully train (for more than 200k steps) a series of new, i.e. randomly initialized, 410M, 6B, 13B, and 52B sized models with the optimal hyper-parameters inferred from the previously mentioned 6B experiments, and observe minimal PTQ degradation in the evaluation suite. For example, across HellaSwag, PIQA, WinoGrande, and LAMBADA, our fully trained 52B model achieves negligible mean degradation while the similarly sized OPT-66B shows ~42% degradation.
>
> > The authors focus on network retraining for improved OPT quantization. For finetuning many approaches have been developed to improve the weight distribution (such as temperature scaling) Do the authors have an idea of how helpful that would for quantization if applied during finetuning?
>
> In our work we completely focused on the impact of pretraining optimization choices on post-training quantization without any additional step such as fine-tuning. However, we agree that measuring the impacts of optimization choices during the fine-tuning step will be an interesting future work. We think that high regularization may have a degree of impact similar to the methods that improve weight distributions. We will add this discussion to the future work in the final manuscript.

---

> > ### Author Response · Authors · 2023-08-16
> >
> > Dear reviewer **vYyu**, we wanted to ask if there are any points that require further clarification. We hope that our response coupled with the additional experiment conducted in the rebuttal phase that show the transferribility of our insights on the effects of pre-training optimization choices on PTQ, to architectures with different types of positional embeddings, address your concern about the transferribility of our method. We also hope that we were able to clarify the significant effect of the choice of fp16 vs bf16 on PTQ.
> >
> > If so, we ask that you consider raising your score to reflect and please do let us know if anything requires further clarification.

---

### Official Review · Reviewer_egL3 · 2023-07-28

**Soundness:** 3 good
**Presentation:** 3 good
**Contribution:** 3 good
**Rating:** 5
**Confidence:** 3

**Summary:**


The authors study the question of whether quantization (of weights and
activations in decoder-only language transformer architectures) fundamentally
reduces performance at large parameter counts, or whether this degradation of
performance, observed in prior works on this topic, is simply a consequence of
certain design choices in the optimizer. They consider post-training
quantization (PTQ) schemes, which simply quantize the weights and activations
of a pretrained (half-precision) model at inference time (to int8/int8) without any extra
quantization-aware retraining, and conduct experiments at the 6B parameter
scale (which entail measuring the accuracy drop after PTQ on 8 datasets)
varying the magnitude of weight decay, dropout rate, gradient clipping, and
half-precision type (fp16 vs. bf16). Their results suggest that larger weight
decay, gradient clipping, and bf16 data types improve PTQ performance; they
then train with appropriate settings at the 60B parameter level, and
demonstrate that they can significantly improve the average PTQ performance
relative to a comparable scale model (OPT-66B); Figure 1 highlights these
results. The authors then study the weights and activations in the OPT-66B
model and their own model in order to assess how their design choices have
affected various properties that played a role in prior narratives on why PTQ
performance might fundamentally be limited at large scales (specifically, the
role of "outlier dimensions", but also general quantization-type metrics such
as reconstruction RMSE).



**Strengths:**

The authors' insights lead to a significant improvement over OPT-66B PTQ
accuracy drop with a comparable parameter count (the accuracy actually improves
on average).  Their results also highlight useful general takeaways about how
regularization can help mitigate PTQ performance drop, including the utility of
bf16, weight decay, and gradient clipping, and they provide reasonable
explanations for why.

The authors have presented most methodological choices and details of their
experimental setup very clearly, which will aid reproducibility of their
results and the ability of researchers to build off the results of their study.



**Weaknesses:**

It might be helpful to include additional quantitative comparisons to prior
work; currently the comparisons to prior art are substantial but mostly
qualitative. iT seems one of the cited works in the related work section,
FlexGen, achieves INT4 quantization of OPT-175B on Lambada and WikiText with
negligible performance (acc/ppl) penalty.  It might also be nice to add brief
experimental studies demonstrating improved throughput/memory footprint of the
authors' PTQ approach, to substantiate the claims made at a few points in the
paper that the authors' approach will have such benefits over the
mixed-precision approach of Dettmers et al.

The description of the experimental protocol for section 4 is somewhat confusing -- lines
145-150 and 193-198 both suggest that training was done using early checkpoints
on 410M parameter models before "scaling" to 6B models, but no results seem to
be reported for 410M parameter models. It seems that this detail could be
omitted if no results of substance are reported to corroborate it. In addition,
it is not made completely clear in section 3.2 whether the 6B experiments
reported in section 4 are actually done via training-from-scratch or done from
early checkpoints; the text in lines 145-150 suggests that section 4 results
are not actually trained from scratch, but the actual protocol in section 3.2
does not clearly state this.

The description of the regularization study in the contributions section feels
slightly misstated -- the "controlled large-scale study" alluded to does not
actually seem to "train from random initialization" (uses early checkpoints),
and the hyperparameter search is actually only conducted at 6B parameter models
(from early checkpoints). This suggests an implicit hypothesis of this study is
that performance differences observed at "medium-scale" models are predictive
of those at larger scales; the fact that this is borne out up to 52B parameter
models seems to be an interesting consequence of the work, but then it does not
seem completely accurate to claim a systematic study up to this scale as a
contribution.

The takeaways from the weight/activation studies in section 5 do not seem to
lend themselves to straightforward conclusions or insights into why the
authors' design choices lead to superior PTQ robustness (see discussion at line
316). It might help the presentation if the authors selected a small subset of
the activation/weight metrics presented which lend themselves to a coherent
interpretation of why the authors' model and OPT/BLOOM have different PTQ
performance. One key takeaway that is mentioned both in the introduction and
the conclusion, that the authors' work refutes the hypothesis that outliers are
not inherent to scaling large language models, is not even presented in the
main body (it has been relegated to appendix B.3), and lines 323-324 suggest
the authors could not find correlations between outlier detection recipes and
PTQ performance. Discussing this study more in section 5 would seem necessary
given the way the authors have presented it as a contribution.


**Questions:**

In line 268, performance differences on BLOOM are claimed; are these results
available or reported? From the appendix Table 4, it seems that these results
are referring to int4 quantization of weights only?

The authors do not test their design choices at the full 175B scale, and the
reason for this is not discussed. It would be helpful to know whether this is
just due to compute obstructions to training at this scale, or whether it is
because the authors' methods are not scaling properly to this scale.



**Limitations:**

Limitations are clearly specified throughout and in the conclusion; there is perhaps somewhat of a missed opportunity to discuss potential negative societal impact of LLM quantization.

---

> ### Author Rebuttal · Authors · 2023-08-09
>
> We thank  **R egL3** for noting that our findings “lead to a significant improvement” and our results “highlight useful general takeaways about how regularization can help mitigate PTQ performance drop”. We are glad they found that our experimental setup is presented “very clearly”, aiding “the ability of researchers to build off the results”.
>
> > It seems one of the cited works in the related work section, FlexGen, achieves INT4 quantization of OPT-175B on Lambada and WikiText with negligible performance (acc/ppl) penalty
>
> Thanks to **R egL3** for this opportunity to clarify the difference between our work and Flexgen. We note that FlexGen compresses only weights but not activations. We are tackling the harder problem of compressing both weights and activations. Thus, it is not comparable with our quantization performance. Conceretly, weights and activation quantization benefits the inference latency of both compute-bound (large batch size,long input sequence and short generation) and memory-bandwidth-bound (small batch size, short input sequence and long generation) use cases while weights only quantization (FlexGen) targets memory-bandwidth-bound use cases only. We will further clarify this difference in the final manuscript and the strength of our contribution, which is currently described briefly in Section 2 and Appendix C.
>
> > an implicit hypothesis of this study is that performance differences observed at "medium-scale" models are predictive of those at larger scales; the fact that this is borne out up to 52B parameter models seems to be an interesting consequence of the work, but then it does not seem completely accurate to claim a systematic study up to this scale as a contribution.
>
> We conduct a large-scale controlled study on the relation between optimization choices (weight decay, gradient clipping, dropout and half-precision training) and quantization robustness with 13 different 6B parameter model variants. While this requires a compute intensive set-up, it allows us to rigorously disentangle what factors actually influence sensitivity to quantization. We then translate the conclusions inferred from the 6B experiments to train models ranging from 410M to 52B parameters from scratch, and validate the robustness of models trained by our recipe. The training of these model variants is incredibly compute intensive (training the 52B model took ~20 days itself) which is why we called this a “large scale” controlled study. We believe this term is justified given the breadth, rigor and computation cost of our experiments.
>
> > lines 145-150 and 193-198 both suggest that training was done using early checkpoints on 410M parameter models before "scaling" to 6B models, but no results seem to be reported for 410M parameter models.
>
> Regarding experiments on 410M, we did the same ablations with 410M models as we did with 6B variants, but omitted the results due to space. We will include them in the final manuscript. We note that we use 6B model experiments (trained from scratch for 75k steps) to infer the best optimization choices because as reported in Dettmers et al., "emergent" quantization sensitivity starts at 6B for the other models, and is representative of larger model sizes as opposed to 410M.
>
> > lines 323-324 suggest the authors could not find correlations between outlier detection recipes and PTQ performance. Discussing this study more in section 5 would seem necessary given the way the authors have presented it as a contribution.
>
> We fully agree with **R egL3** that having a detailed discussion on outlier dimensions earlier in the manuscript would help make our argument more cohesive. We will move the main parts of the detailed outlier analysis from Appendix B.3 to section 5 for the final manuscript.
>
> > It might help the presentation if the authors selected a small subset of the activation/weight metrics presented which lend themselves to a coherent interpretation of why the authors' model and OPT/BLOOM have different PTQ performance
>
> We agree with this suggestion from **R egL3**. We point out that RMSE($X,\widehat{X}$) and input token activation STD strongly correlate with degradation trends we observe in OPT and BLOOM models. Comparing our model with the other two, we look at all the attention projection layers (attn-kqv-proj) across the models and find that RMSE($X,\widehat{X}$) for OPT and BLOOM models is significantly higher than corresponding values in our 6B model. This is depicted in Figure 11, Appendix B.2. More concretely,  RMSE($X,\widehat{X}$) values for OPT-6B and BLOOM-7.1B are up to **33.2x** and **13x** the corresponding values for our fully trained 6B model, respectively. A similar pattern exists for STD values which are consistently higher for OPT-6B and BLOOM-7.1B than that of our model, up to 3.6x and 2x, respectively.
>
> Based on **R egL3’s** suggestion, we will put more emphasis on these metrics and make more clear that RMSE and STD explain the OPT & BLOOM degradation patterns in the final version of the paper.
>
> > In line 268, performance differences on BLOOM are claimed; are these results available or reported? From the appendix Table 4, it seems that these results are referring to int4 quantization of weights only?
>
> The numbers in lines 266-273 are regarding INT4 weight-only column-wise quantization. We compare against the INT4 BLOOM results reported in the GLM-130B technical report in Table 10. We will make this source clear in the final manuscript.
>
> > The authors do not test their design choices at the full 175B scale, and the reason for this is not discussed
>
> This was purely due to the enormous cost of fully training a 175B parameter model given all our model variants are trained from random initialization. We note that the 52B model required training time of roughly 20 days and in total we trained 17 models of sizes ranging from 410M to 52B to carefully measure the impact of each ablation. We will add this explanation to the final version of the paper.

---

> > ### Author Response · Authors · 2023-08-14
> >
> > Now that discussion is underway, we wanted to ask **R egL3** if there are any follow up points we can clarify. If there are no further points of clarification regarding the manuscript and the numerical efficiency benchmarking experiments conducted during the rebuttal period demonstrating the improved latency, throughput, and GPU memory usage of our INT8 recipe, we would ask that reviewer **egL3** to consider increasing their score to reflect.

---

### Author Rebuttal · Authors · 2023-08-09

We greatly appreciate the thoughtful and positive feedback from the reviewers. We are encouraged that reviewers found the results notable [xK34, egL3] “insights lead to a significant improvement over OPT-66B PTQ accuracy”. Reviewers also positively reviewed our experimental setup as rigorous, clear and comprehensive [xK34, egL3, vYyu] “The study rigorously analyzed the impact of optimization choices on sensitivity to quantization in models ranging from 410 million to 52 billion parameters.” We are also glad that the reviewers found the problem we try address to be well motivated  [TcCw, vYyu] “Related work strongly suggests [to me] the difficulty of quantizing transformer-based language models, in particular when both the network weights and activations are quantized.”, and the paper to be “clearly written” and “easy to follow” [TcCw, egL3, vYyu].

We are heartened by the reviewers' acknowledgment of the potential brought by our work for opening up avenues for more efficient quantization methods, raising questions about the inherent nature of other emergent properties “the authors demonstrated that reoccurring activation outliers are not a universal emergent property of large language models at scale”  and “provides a detailed analysis of activations and weights, offering insights into the key characteristics influencing sensitivity to quantization” [xK34, egL3, vYyu].

Below we take this opportunity to clarify some of the shared concerns between reviewers:


**1. On the transferability of our results to the OPT family and models with different architectures**

[vYyu] and [TcCw] raised valuable points asking how transferable our insights are to other model families such as OPT.  For clarity, all the models that we compare to in the paper including OPT and BLOOM share the same GPT-style architecture.

All the models that we experimented with are architecturally very similar to the OPT family as they are both decoder-only, pre-layernorm (where the layernorm is placed before the linear layers), use vanilla attention, and use learned positional embeddings. They differ in the number of blocks and the hidden dimensions of models at different sizes. Since we thoroughly experimented with different model sizes, we believe that our insights directly translate to the OPT family’s architecture.

However, we realize that the type of positional embedding is often a differentiating factor between architectures e.g. OPT models use learned embeddings (same as us) whereas the BLOOM family uses ALiBi. As a result, in the rebuttal phase, we trained 3 models to 75k steps using learned and sinusoidal positional embeddings, and ALiBi. We used otherwise optimal hyper-parameters (bf16 training, weight decay=0.1, dropout=0.0, gradient clipping=1.0). We did not see a notable difference in downstream PTQ degradation as all values were within 0.06% of another.

Therefore, we believe that it is regularization choices rather than small architecture differences, significantly impact models’ PTQ performance. In support of this, our fully trained 52B model shows 0.0% degradation across a set of 4 tasks whereas a similarly sized OPT-66B model shows a 42% drop, confirming this impact.


**2. Clarifications on our experimental setup**

As raised by [egL3], we would like to provide further clarification on the details of our training and experimental setup. By the term “early checkpoints”, we do not mean fine-tuning or training from pre-trained checkpoints, but we refer to training from scratch and stopping relatively early at 75k steps (instead of training fully for more than 200k steps). Therefore, we do not perform any form of fine-tuning from pre-trained checkpoints on any of the models reported in the paper. We would like to clarify that **all of our models including the 13 early checkpoint 6B models, and the fully trained 410M, 6B, 13B, and 52B models, are trained from scratch (from random initialization).**  This is much more computationally expensive, but we believe this is critical to perform a controlled ablation of each variant without confounding factors like different pre-training conditions. We will make this clarification explicitly in the manuscript.


**3. Numerical results on improved efficiency of our recipe**

We thank [egL3] for suggesting that we include “improved throughput/memory footprint of our PTQ approach”. We take this opportunity during rebuttals to run additional benchmarking experiments to act on this feedback. Concretely, we benchmark end-to-end inference speed for our PTQ recipe on our 52B model for both compute-bound (large batch size, long input sequence and short generation) and memory-bandwidth-bound (small batch size, short input sequence and long generation) use cases. We use standard CUDA kernels that enable INT8 matrix multiplication. Across all settings, we find that our vector-wise PTQ approach provides a 1.4-1.5x latency speedup (or 1.4-1.5x throughput improvement) and around 40% memory footprint reduction over half-precision inference. We note that LLM.int8() is slightly slower than half-precision inference (Dettmers et. al.). Therefore, our PTQ approach is significantly faster than LLM.int8(). Additionally, Dettmers et. al. also show that LLM.int8() leads slower matrix multiplication compared to vector-wise PTQ. We include the inference speed benchmark table in the attached pdf for reference with further details and will include it in the final manuscript.


We are grateful for the valuable insights provided by the reviewers, and we are dedicated to addressing the feedback and incorporating it into the final version of our paper to further enhance its quality and contribution.

---

### Decision · Program_Chairs · 2023-09-21

**Decision:**

Accept (poster)

**Comment:**

This paper studies quantization in LLMs, specifically it investigates the drop in performance which usually accompanies quantization of large-scale models. The effect of quantization is currently unclear in the literature, with some models retaining their performance much better than other models of similar size. The authors study show that optimization parameters strongly affect the post-training-quantization (PTQ) performance, and use their insights to pick better hyperparameters for certain models — showing that these large models can in fact be quantized at scale, given the right optimization.


Reviewers agree that the paper is well-written, the topic is important, and the experiments are thorough and largely convincing. Thus, I recommend acceptance.
I encourage the authors to address the reviewers' comments in the camera-ready, which will result in a stronger paper. In particular, Reviewer egL3 notes several instances where the claims as phrased may not be exactly supported by the evidence. I encourage the authors to revise this language.